# Optogenetics inspired transition metal dichalcogenide neuristors for in-memory deep recurrent neural networks

Rohit Abraham John [1], Jyotibdha Acharya[2,3], Chao Zhu [1], Abhijith Surendran[2], Sumon Kumar Bose [2], Apoorva Chaturvedi[1], Nidhi Tiwari[4], Yang Gao[1], Yongmin He [1], Keke K. Zhang[1], Manzhang Xu [1], Wei Lin Leong [2], Zheng Liu [1], Arindam Basu [2✉] & Nripan Mathews [1,4✉]

Shallow feed-forward networks are incapable of addressing complex tasks such as natural language processing that require learning of temporal signals. To address these requirements, we need deep neuromorphic architectures with recurrent connections such as deep recurrent neural networks. However, the training of such networks demand very high precision of weights, excellent conductance linearity and low write-noise- not satisfied by current memristive implementations. Inspired from optogenetics, here we report a neuromorphic computing platform comprised of photo-excitable neuristors capable of in-memory computations across 980 addressable states with a high signal-to-noise ratio of 77. The large linear dynamic range, low write noise and selective excitability allows high fidelity opto-electronic transfer of weights with a two-shot write scheme, while electrical in-memory inference provides energy efficiency. This method enables implementing a memristive deep recurrent neural network with twelve trainable layers with more than a million parameters to recognize spoken commands with >90% accuracy.

[1] School of Materials Science and Engineering, Nanyang Technological University, 50 Nanyang Avenue, Singapore 639798, Singapore. [2] School of Electrical and Electronic Engineering, Nanyang Technological University, 50 Nanyang Avenue, Singapore 639798, Singapore. [3] HealthTech NTU, Interdisciplinary Graduate Programme, Nanyang Technological University, Singapore 637335, Singapore. [4] Energy Research Institute @ NTU (ERI@N), Nanyang Technological University, Singapore 637553, Singapore. ✉email: arindam.basu@ntu.edu.sg; Nripan@ntu.edu.sg

The success of deep learning in diverse fields such as image classification[1] and face recognition[2] has spurred a renewed interest in the area of artificial intelligence (AI). Despite the impressive progress already demonstrated with conventional CMOS-based programmable architectures[3], innovative neuromorphic hardware approaches are required to emulate the scale, connectivity and energy efficiency of biological neural networks[4,5]. The first wave of neuromorphic hardware solutions based on electrical memristors have demonstrated advantages in parallel summing and update operations, scalability, cost, and power consumption[6–8]. However, the switching dynamics of conventional memristors produces abrupt transitions and write non-linearity, resulting in asymmetric weight updates and limited number of accessible states[9]. Such limited precision weights can only be used for shallow feed-forward networks trained to classify simple datasets and are incapable of addressing applications like speech recognition and natural language processing (NLP) that require learning of temporal signals. To address these requirements, we therefore need neuromorphic architectures with recurrent connections and deeper architectures (>10 hidden layers) such as deep recurrent neural networks (DRNNs)[10,11]. However, the efficient training of DRNNs calls for fully-parallel write/read operations, which demands non-volatile memory elements with multiple linearly distributed conductance states addressable via blind updates- not satisfied by current electrical implementations. A potential way to accelerate the training process of DRNN is to exploit the increased speed offered by optical processing platforms. However, such all-photonics-based computing platforms require meticulous integration of numerous optical components with a large footprint[12] and are incapable of updating weights in a precise, linear manner, necessary for training DRNNs via parallel updates[13]. Present electrical and optical approaches therefore do not sufficiently address the effective bit precision and accuracy required for energy-efficient computation-in-memory implementation of DRNNs.

This motivates us to re-examine neuroscience, specifically those based on optogenetics, for a second wave of neuromorphic devices for advanced AI applications[14]. Optogenetics- a photo-stimulated neuromodulation technique, utilises optical pulses to monitor and manipulate neuronal activities by controlling ionic currents in biological tissues[15,16]. With higher speed and spatiotemporal precision over its electrical counterparts, optogenetics has facilitated precise probing of neuronal circuits[14], unravelling underpinnings of cognition and memory. Beyond probing the neural pathways, optogenetics is now extensively employed to stimulate and silence neural activity to manipulate locomotion[17] and rewire neural pathways to cure disorders[18]. This strategy of utilising light to tune the learning and memory behaviours[19,20] can be adopted to selectively activate artificial neurons and synapses in hardware circuitry to address the drawbacks of limited number of accessible states, non-linearity and asymmetric weight updates. Inspired from optogenetics, we propose an optoelectronic neuromorphic computing platform comprising of photo-excitable neuristors (PENs) to selectively control the excitability of artificial neurons and update the weights of synapses with high linearity to obtain precise weights necessary for DRNNs.

Since we derive our inspiration from optogenetics, we first demonstrate light-modulated firing of neuron circuits and spike-timing dependent plasticity (STDP)-based synapses on atomically-thin low-bandgap PENs with rhenium disulfide (ReS$_2$) active channels (Supplementary Note 1, Supplementary Fig. 1), to benchmark our devices against state-of-the-art spike-based learning synapses. We illustrate spatiotemporally selective perturbation: a key factor in optogenetics, to demonstrate the possibility of selectively probing a specific group of neurons/synapses

with high precision. To demonstrate the excellent properties of our devices, we extensively characterise the switching characteristics and calculate the endurance, cyclability, signal-to-noise ratio (SNR) and linear dynamic range (LDR) for benchmarking against state-of-the-art. The reported number of accessible non-volatile states (980 or ~10-bit equivalent) is the highest till date to the best of our knowledge. The SNR values as high as 77 are also among the highest reported till date. Since the internal adjustment of weights in a neural network can be simplistically viewed as mathematical addition and subtraction operations or their extensions, we demonstrate abacus operations to illustrate the weight changes of synaptic connections in a more tangible and comprehendible manner. Finally, we propose a new method of optical writing to transfer offline learnt weights on to the device, after which the device is used for inference in electrical mode only without requiring optical inputs, resulting in high energy efficiency due to in-memory computing. Offline learning enables DRNN training with advanced learning rules while the exquisite write linearity afforded by the optical gating is the major phenomenon we exploit to get very accurate weight transfer with a two-shot write scheme. The proposed PEN features an order of magnitude higher LDR than other recent state-of-the-art reports, enabling us to simulate a DRNN for speech recognition with an order of magnitude higher parameters than digit recognition networks.

## Results

**Inspiration from optogenetics.** While conventional optogenetics targets specific neurons for excitation or inhibition, our approach of using PENs gives us the flexibility to selectively activate either neurons or synapses in an artificial neural network (Fig. 1a). We first demonstrate the selective ability to address neurons, a hallmark of optogenetics. The PENs were configured to construct an integrate and fire (I&F) circuit as shown in Fig. 1b and Supplementary Note 2, Supplementary Table 1. Upon illumination of the PEN, the conductance increased, resulting in higher firing rate of the I&F neuron in contrast to a photo-insensitive one (Fig. 1b), in direct correlation to conventional optogenetics. This selective neuronal photo-excitability is hitherto not demonstrated for hardware neuromorphic circuits. To qualify as optoelectronic synapses, the PENs were next benchmarked on the degree of manipulation of spike-timing-dependent plasticity (STDP) windows as a function of the optical stimuli. STDP is a figure-of-merit of plasticity of memristive elements for unsupervised learning[21,22]. Asynchronous electrical spikes of identical amplitude and duration induced a standard asymmetric Hebbian STDP function in the PENs (Supplementary Note 3, Supplementary Fig. 2). Illumination induced a permanent increase in the device conductance or long-term potentiation (LTP), resulting in a positive shift of the electrical STDP functions (Fig. 1c). Increasing intensity or energy of illumination resulted in larger positive shifts of the electrical STDP windows, resulting in an output dependent on both electrical and optical inputs (Supplementary Note 3, Supplementary Figs. 3 and 4). This modulation of STDP windows with light or photo-modulated STDP resemble biological optogenetic measurements[23,24], qualifying these PENs as equivalents to optogenetic actuators or opsins. In all the above measurements, programming weights with optical pulses resulted in near-linear changes of the electrical STDP window as a function of the dosage (wavelength and intensity) of incident light. This opens up the possibility of accessing linearly distributed weights with high precision utilizing optical blind updates, essential for DRNNs.

Spatiotemporally selective perturbation is another key factor in optogenetics, enabling probing of a specific group of neurons with

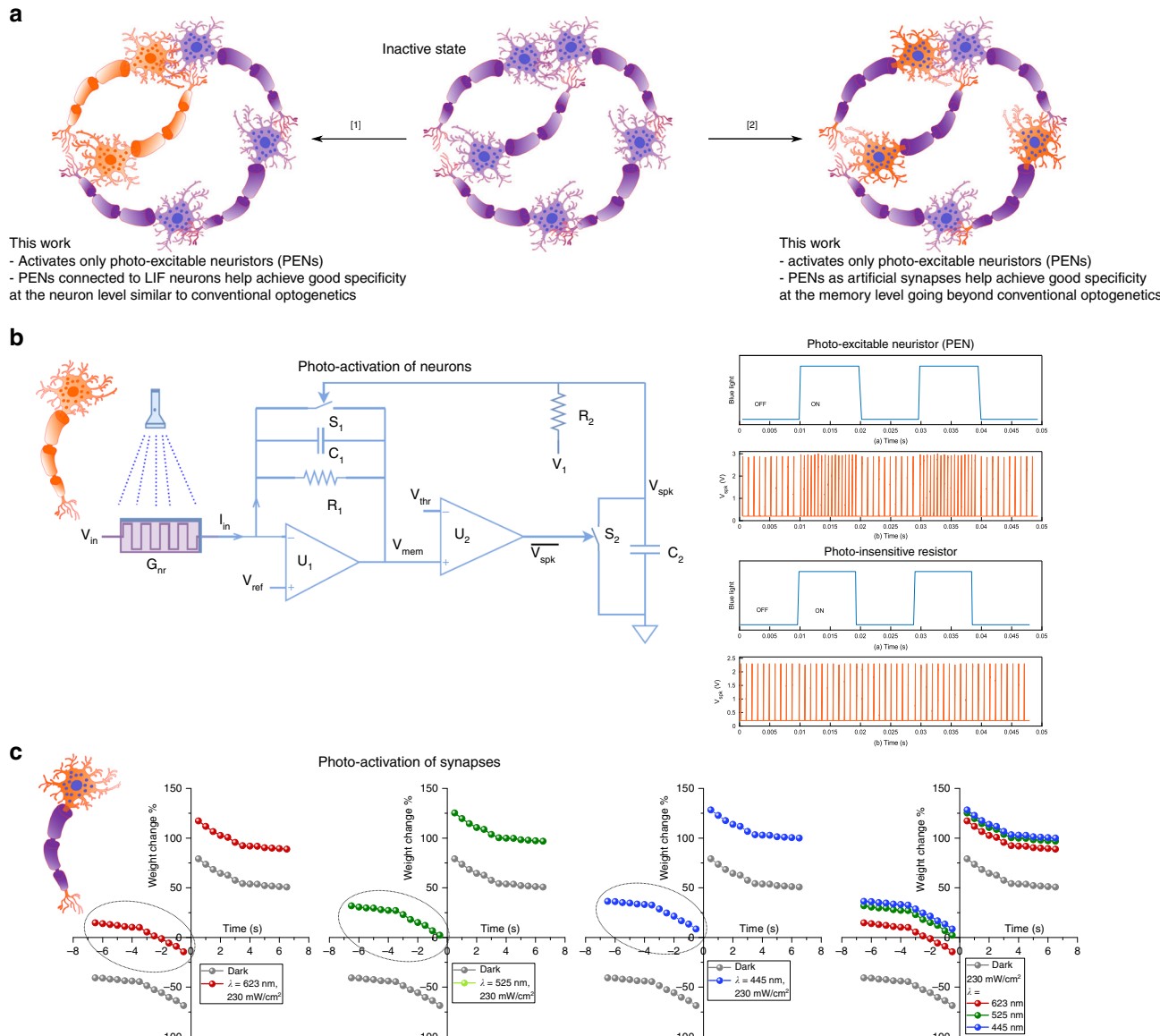

**Fig. 1 Optogenetics-inspired light-driven computational circuits. a** Inspiration from optogenetics. Our approach of using PENs in an artificial neural network gives us the flexibility to selectively activate either neurons or synapses. **b** An I&F circuit is designed using two operational amplifiers and the PEN functionally mimicking optically active ion-channels in optogenetics. Without light activation, the neuron fires at a baseline rate set by the reversal potential $V_{in}$ of the artificial ion-channel. On illumination by blue light ($\lambda = 445$ nm, amplitude = 5 mWcm$^{-2}$ and pulse width = 10 ms), the conductance of the PEN ($G_{nr}$) increases, resulting in a higher frequency of neuronal firing (top). An I&F neuron constructed using conventional resistors does not show any change in firing rate on optical stimulation (bottom). **c** Photo-modulated spike-timing-dependent plasticity learning rules in PENs. The graphs (read row-wise) depict the modulations of STDP upon illumination with red (623 nm), green (525 nm) and blue (445 nm) light with an intensity = 230 mWcm$^{-2}$.

high precision. To demonstrate selective perturbation, a $6 \times 6$ mini-array of PENs was selectively subjected to sinusoidal global clocks of red and blue wavelengths (Fig. 2a). The PENs subjected to blue irradiation resulted in a larger modulation of the synaptic weights with the final spatial conductance map corresponding to ~102 nS, while those activated by red light depicted a final conductance ~13 nS. This difference in photoresponse or wavelength selectivity allows a wide flexibility for setting the threshold for neuronal firing (Supplementary Note 3, Supplementary Fig. 5). To substantiate selective activation further, a 36-element array comprising of 16 light-sensitive ReS$_2$ and 20 insensitive indium tungsten oxide (IWO) synapses was assembled as shown in Fig. 2b. Details of the IWO transistor fabrication and synaptic characterisation is detailed in our earlier work[25]. All synapses were initially programmed to a common low

conductance state (5 nS). Upon illumination with red light ($\lambda = 623$ nm, 65 mWcm$^{-2}$ intensity), the channel conductance of the ReS$_2$ PENs depicted a non-volatile increment (5–6.5 nS) due to its low bandgap (1.5–1.8 eV)[26]. On the other hand, the channel conductance of IWO synapses remained constant (5 nS) due to their large bandgap (3.6 eV)[27]. A spatial conductance map readout of the array resembled the image of "N" with the light-sensitive ReS$_2$ PENs spatially arranged to resemble this alphabet as shown in Fig. 2b. These results illustrate the high spatial specificity of the light-based write operations, globally capable of addressing light-sensitive neural elements on demand. By employing semiconductors of varying band-gap and optical absorption, this concept can be extended to a wide variety of material systems and wavelength division multiplexing schemes, enabling highly selective probing of artificial neural networks.

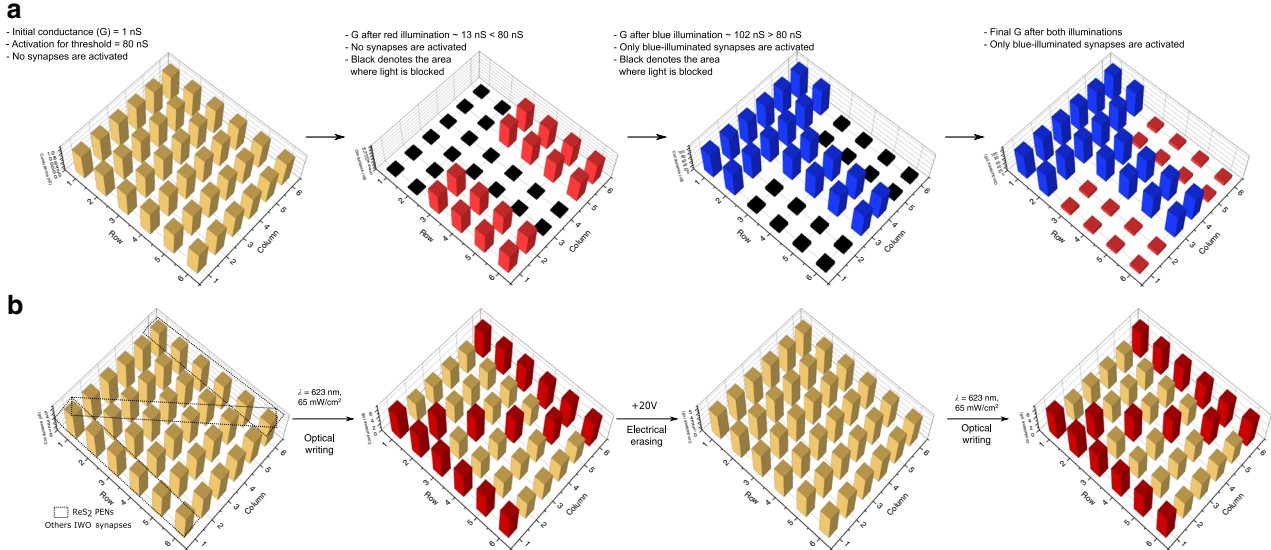

**Fig. 2 Spatiotemporally selective perturbation. a** A 36-element array of PENs was assembled and initially programmed to a common low conductance state (1 nS). Illumination with optical sinusoidal stimuli (red and blue) selectively activated the PENs with each device depicting a non-volatile weight change corresponding to the wavelength of illumination. Blue illumination triggered significant weight changes in the devices resulting in a final spatial conductance map ~102 nS, crossing the pre-set activation/firing threshold of 80 nS. On the other hand, red wavelength stimulation triggered weight changes to a final value of ~13 nS, inactivating these synapses. **b** A 36-element array comprising of 16 light-sensitive ReS$_2$ and 20 insensitive IWO synapses was assembled and initially programmed to a common low conductance state (5 nS). Spatial conductance maps of the synapses when subjected to red ($\lambda = 623$ nm) optical stimuli (intensity = 65 mWcm$^{-2}$) is shown. Illumination with optical stimuli selectively activated the light-sensitive low-band-gap ReS$_2$ PENs without affecting the light-insensitive high-band-gap IWO synapses. A spatial conductance map readout of the array resembled the image of "N" with the light-sensitive ReS$_2$ PENs spatially arranged accordingly as shown.

**Optoelectronic switching for deep recurrent neural networks**. From the algorithmic point of view, an ideal synapse should typically satisfy linear weight updates for high classification accuracies using online blind programming schemes[28]. Even for transferring offline learnt weights with high bit precision to a neuromorphic hardware necessary for speech recognition or NLP, blind weight updates without iterative read-write procedures are necessary to avoid prohibitively long write times for large-scale DRNNs[29]. Update linearity and write noise have been identified as the prime analogue device properties which degrade the accuracy of neural networks[9,30]. However, most memristive systems depict a nonlinear and asymmetric response due to their abrupt filamentary switching physics, failing to sufficiently address the bit precision and accuracy required for in-memory computations in DRNNs. We propose to exploit the exquisite write linearity and low write noise afforded by the optical gating to get very accurate weight transfer with a two-shot write scheme.

To demonstrate the extremely high density and linearity of non-volatile states available for analogue computation, the PENs were subjected to an input optical pulse train of constant pulse width and interval. Upon illumination ($\lambda = 623$ nm, 65 mWcm$^{-2}$ intensity), the conductance updates exhibited excellent linearity, non-volatility and retention characteristics. Real-time monitoring of the conductance changes revealed a precise stepped linear increase in conductance from 1.25 to 180 nS with a step size of ~0.18 nS, equivalent to 980 distinct conductance states (~10-bit) (Supplementary Note 4, Fig. 6). Shorter activation wavelengths resulted in higher slopes, wider conductance ranges, higher retention and SNR (Fig. 3a, Supplementary Note 4, Supplementary Figs. 7 and 8). Figure 3a depicts a magnified view of the conductance transitions (states 1–16 and 965–980) upon blue illumination ($\lambda = 445$ nm, 65 mWcm$^{-2}$ intensity), while Supplementary Note 4 and Supplementary Fig. 8 depicts all the 980 conductance transitions achieved with the PEN.

An ideal neuromorphic device should depict a narrow distribution centred about the average change in conductance due to a write operation ($\Delta G_{avg}$) over the entire range of conductance ($G$) to enable a high dynamic range of weights with extremely accurate weight updates[31]. A test of write endurance (1600 switching cycles – 800 write and 800 erase steps between the 1st 16 conductance states) revealed high endurance and low write noise in the investigated devices for all wavelengths of optical stimulation (Fig. 3b–j). Supplementary Note 5 and Supplementary Fig. 9 depicts a magnified view of the conductance response during the first 128 voltage-controlled write and erase operations and limits of linear weight updates, and Supplementary Fig. 10 depicts the endurance of the non-volatile states. The conductance changes exhibited predictable, saw-tooth-like conductance steps without any degradation in the overall channel conductance. Probability distribution plots of the write-erase processes depicted exceptionally high write-linearity with an average SNR ($\Delta G_{avg}^2/\sigma^2$; $\sigma$- standard deviation) of 39 (potentiation)/30 (depression), 77 (potentiation)/24 (depression) and 75 (potentiation)/53 (depression) for red, green and blue wavelength optical write and their corresponding electrical erase operations (Fig. 3e–j). This is among the highest SNR reported till date and the first demonstration of such linear weight updates with optical stimuli. In comparison, state-of-the-art filamentary RRAMs and PCMs showed lower SNR (<1) and orders of magnitude greater non-linearity[28].

To demonstrate the precise weight-update capability of our PENs through blind optical writes and electrical erases in a tangible form, we implement an optoelectronic arithmetic calculator analogous to an abacus that requires highly linear operations for summing and subtraction (Fig. 3k, Supplementary Note 6, Fig. 11). To accommodate multi-digit calculations, two PENs were used – one (blue) to represent the first and the other (red) to represent the second digit of the base-10 number system. Both devices were pre-programmed to a common low

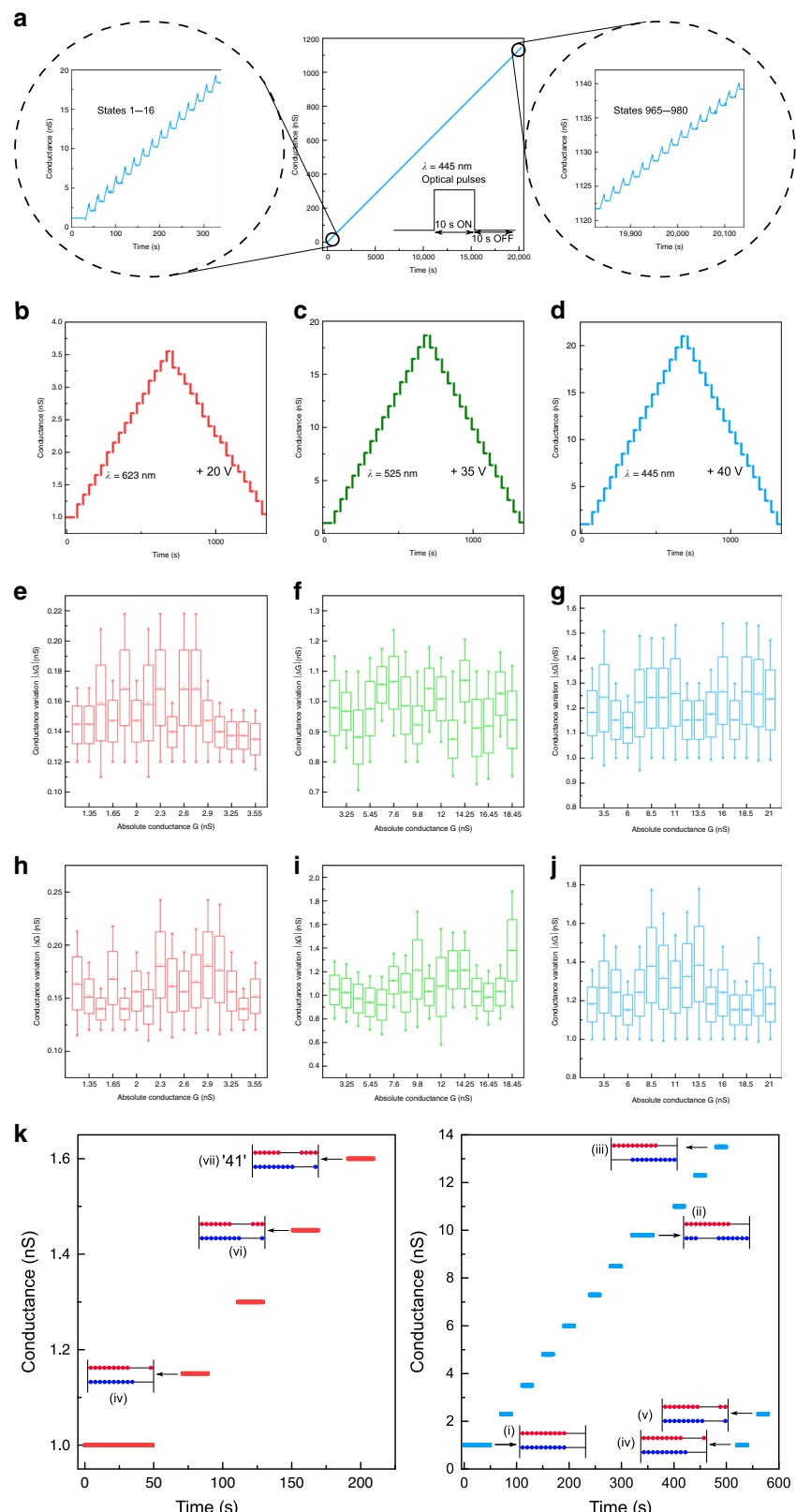

conductance state (1 nS, representing '0') prior to any calculations (stage-i). In abacus, this is represented by an extreme left shift of all the beads. For an addition operation of '27 + 14' (Fig. 3k), the 1st PEN was optically programmed with as many steps as the augend ('7' in this case) (stage-ii). In abacus, this is represented by right shift of an appropriate number of blue beads. This was

followed by programming steps equivalent to the addend '4'. When the sum reached '10' (stage-iii), the unit's place PEN was RESET to the initial conductance state by appropriate electrical pulses (amplitude = +40 V, pulse width = 1 s, number = 10, Supplementary Note 5, Fig. 9a), represented by extreme left shift of all the blue beads (stage-iv). Concurrently, the second PEN

**Fig. 3 Optically addressable multi-level memory for DRNNs.** An input optical pulse train of constant pulse width and interval (10 s, 65 mWcm$^{-2}$) intensity resulted in a precise stepped increase in conductance equivalent to 980 distinct conductance states. **a** shows a magnified version of the conductance states 1–16 and 965–980 activated by 445-nm wavelength of optical stimuli. To maximise the dynamic range of linear response, the devices were electrically pre-programmed at −40 V to ensure a low initial conductance state. This electrical bias was also maintained constant throughout the optical writing steps. Analysis of endurance during write-erase operations. Application of synergistic optoelectronic pulses result in precisely controlled near-symmetric bidirectional weight changes via optical potentiation and electrical depression as shown in **b–d**. Positive weight changes activated by 623-nm optical stimuli were erased by electrical pulses of amplitude +20 V. Higher electrical voltages were applied at the back gate to erase the higher degree of potentiation induced by shorter wavelength optical stimulation. For instance, positive weight changes of higher magnitude due to 525/445 nm optical activation were erased by electrical pulses of amplitude +35/40 V respectively. **e–j** ΔG vs G box and whisker plots – a measure of endurance and write noise during cycling tests. The box portion of the box plot are defined by two lines at the 25th and 75th percentiles and the line drawn inside the box represents the median (50th percentile). Plots **e–g** depict the variation in the changes in conductance during the optical write operations, while **h–j** depict the variations during the electrical erase operations for all the three wavelengths of optical stimuli. **k** Optoelectronic abacus operation. Precise optical potentiation and electrical depression enabled facile emulation of arithmetic operations, analogous to an abacus. Multiple PENs were employed to represent the unit's (blue bead) and ten's (red bead) place, and programming steps were designed as per the arithmetic operation under calculation. Optical stimulations resulted in potentiation represented by the rightward sliding of the beads, while electrical stimulations caused depression (leftward sliding of the beads). The methodology of the arithmetic operations are indicated with necessary illustrations as insets.

representing the ten's place was updated by a single programming pulse (carry over) to hold '1' temporarily (stage-iv). In abacus, this carry over operation is represented by a right shift of a single red bead. Subsequently, programming steps equivalent to the remainder ('1') was carried out to get a final sum of '1' in the unit's place (stage-v). Similar operations were carried out on the second PEN (currently holding '1' from the carry over) resulting in a final sum of '4' in the unit's place (stages-vi–vii). The conductance states of both PENs were read by a reading voltage of 0.1 V to get the result '41'. Detailed illustration of other arithmetic operations are provided in Supplementary Note 6 Supplementary Fig. 11. The in-memory computing capabilities elementarily demonstrated as arithmetic operations, represent the weight update protocols underlying powerful approaches like backpropagation in neural networks.

The linear increase/decrease of weights in a material system depend on a number of factors including the nature of traps and its temporal response, the kinetics of photo-carrier generation and their recombination. The programming scheme we adopt caters for many of these factors (Supplementary Note 7, Supplementary Figs. 12–16 and Supplementary Note 9, Supplementary Figs. 22–25). We attribute the optical modulation due to persistent photoconductivity (PPC)[32] to be the reason for linear programming of weights in our work. Upon illumination or in other words during each optical write operation, photocarriers (electrons) generated in the channel increase the conductance (potentiation) and the photogenerated holes are localised/trapped in states within the semiconductor bulk or/and at the semiconductor-dielectric interface resulting in a delayed recombination. This photogating effect[33] results in a permanent change in the channel's carrier concentration, resulting in a non-volatile weight update. Carriers generated during the subsequent illumination adds to the existing carrier concentration with every input pulse, updating weights in a linear manner. While optical pulses enable linear incremental non-volatile write steps, electrical gating is used to erase via defect-assisted recombination. During erase, the electrical gating raises the Fermi level to induce electron accumulation in the channel and this accelerates the recombination with the holes in the trap sites reducing the photoconductance[34] (depression) as shown in Fig. 3b–d and Supplementary Note 5, Fig. 9. The number of distinct states are determined primarily by the programming pulse resolution and recombination kinetics of the photo-generated carriers, and hence, the programming pulses could be chosen to achieve a near-perfect linearity. Detailed explanation on the mechanistic understanding, choice of gate and drain voltages, and the

illumination conditions to maximise the dynamic conductance range of linear weight updates is provided in (Supplementary Note 7, Supplementary Figs. 12–16 and Supplementary Note 9, Supplementary Figs. 22–25).

**Simulations.** The extremely low noise and high linearity in writes was exploited to create high-accuracy energy-efficient neural networks for pattern classification by implementing vector matrix multiply (VMM) operations (that form the bulk of the computations[35]) within memory (Fig. 4a–c, Supplementary Note 8, Supplementary Figs. 17–21). Unlike convolutional neural networks (CNNs)[36–38], DRNNs like long short term memory (LSTM)[10,39] that achieve state of the art results in speech classification, require much more precise weights[11,40]. Earlier work[41,42] on memristive devices have shown their ability to classify handwritten digits from the MNIST dataset[43] using fully connected networks (FCN) with online learning. However, such online learning schemes using stochastic gradient descent are unsuited for training DRNNs on large-scale datasets; instead sophisticated momentum based learning rules such as ADAM[44] are preferred for offline training which enables gradient averaging across mini-batches. Hence, in this work we use offline learning of weights followed by optically-assisted weight transfer to the PEN crossbar which can then perform the inference operation in electrical mode with extremely low energy dissipation (Fig. 4b).

Compared to CNNs, DRNNs require higher weight precision (≥6 bits) due to the recurrent layers[40] (Fig. 4c). Programming weights in non-volatile memories with high precision require iterative read-write schemes with 20-30 iterations being common to account for device non-linearity and variability between multiple memory elements[45–47]. We exploit the exquisitely high linearity and low write noise of the PENs to propose a two-shot write scheme for writing weights with high accuracy. Characterisation of the PENs across five devices and three wavelengths show a high linear dynamic range (LDR > 6000) obtained by dividing the linear range of weights (conductance range where deviation from best-fit straight line is less than half of step size) by the standard deviation of write noise (Fig. 3, Supplementary Note 8, Supplementary Fig. 17). However, measuring all states to get a best-fit line is not scalable to DRNNs with millions of weights and infeasible for reprogramming deployed devices. A practical way to estimate the slope of the straight line is to use a two-shot measurement procedure that eliminates multiple read-write cycles per PEN at the cost of slope estimation errors. A hardware friendly method (adopted for simulations shown) of using global optical potentiation to the maximum weight followed

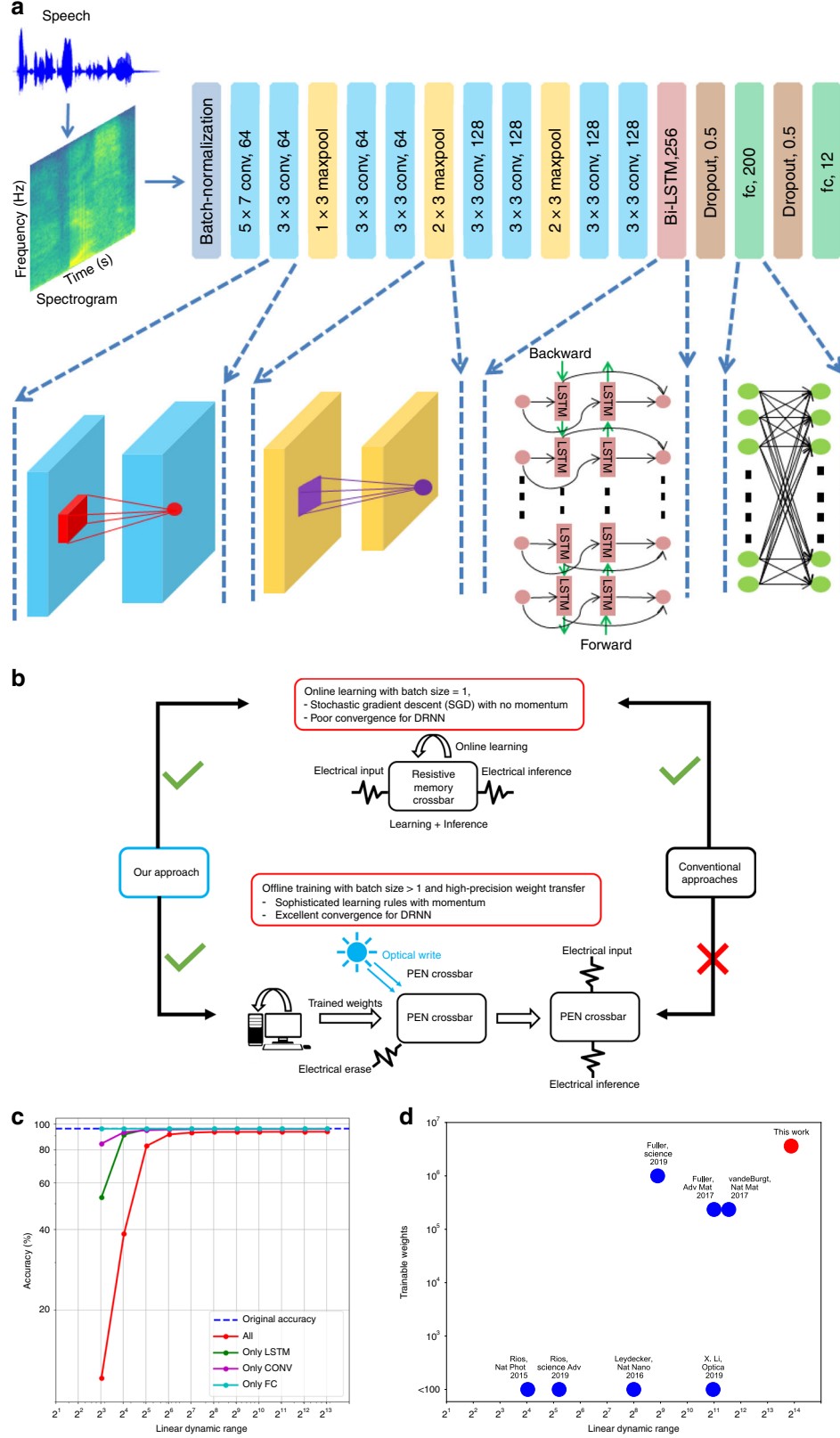

by selective electrical depotentiation required one optical write, two electrical writes and two measurements per PEN (Supplementary Note 8, Supplementary Fig. 18). The linearity of the write operation allows simple one parameter (slope) calibration avoiding more complex read-write schemes for non-linear elements[45–47] and reducing the number of iterations by an order of magnitude. In mapping the weights to conductance values, both write noise and slope estimation errors were considered. While noise sets the lower limit of achievable weight update, estimation errors limit the dynamic range of weight values accessible by our two-shot measurement scheme to ~600 (Supplementary Note 8, Supplementary Fig. 19). The number of

**Fig. 4 Simulation of a DRNN for keyword recognition.** A deep neural network with 12 trainable layers including convolutional, fully connected, and recurrent LSTM layers is used to classify 12 different spoken digits. **a** The detailed architecture of the network is shown with all filter sizes and dimensions mentioned. **b** Highly linear weight update of PENs allow us to transfer high precision weights from offline learnt DRNNs (using sophisticated learning schemes with mini-batch averaging for smooth convergence) along with online learning with blind updates for simpler feed-forward networks. **c** The accuracy obtained in electrical inference with a two-shot opto-electronic write of weights is within 2.5% of the software simulation with floating-point numbers. The linear dynamic range of the PEN is limited to ~600 due to the two-shot write scheme. Simulating only one layer of convolution, fully connected or LSTM layers show the recurrent LSTM layers to be most sensitive to low weight dynamic range. **d** PENs used in our work exhibit an order of magnitude higher linear dynamic range than other recent reports, enabling simulation of DRNNs for speech recognition with an order of magnitude higher weights than other reported simulations for handwritten digit recognition.

write operations could be further reduced by using optical means to set the weights directly to the desired value provided selective optical stimulation is possible.

As a first experiment to prove the efficacy of our two-shot write method and to compare with other reported works, we demonstrate results for classifying digits from the MNIST dataset using a combination of CNN and FCN with four trainable layers. For MNIST classification, the four layer network programmed with the two-shot scheme achieves an accuracy of 99.09%(±0.1) that is almost indistinguishable from its software full precision counterpart (99.2%) and at par with the state-of-the-art[41,48] (Supplementary Note 8, Supplementary Fig. 20). Going beyond such shallow networks, we next demonstrate classification of spoken commands[49] using a twelve trainable layer DNN with convolutional layer, FCN, and bi-directional LSTM layers (Fig. 4a), where the VMM operations in all layers are implemented using PENs. For the speech classification problem, spectrograms of dimension $256 \times 101$ were input to the network. On the twelve class speech classification task, the PEN-based VMM with two-shot writes achieves 93.25%(±0.2) accuracy which is only ~2.5% less than its software counterpart. The LSTM layers are found to be most sensitive to low LDR due to recurrence (Fig. 4c). Without slope estimation error in two-shot updates, the high LDR > 6000 of our PENs help achieve near-perfect accuracy (Supplementary Note 8, Supplementary Fig. 21).

## Discussion

Thus, we demonstrate that such optoelectronic neuromorphic devices can be adapted to execute highly parallel energy-efficient blind weight-update protocols for DRNNs, accelerated by in-memory computing. In comparison to state-of-the-art, the proposed PEN features an order of magnitude higher LDR facilitating an order of magnitude lower iterations for weight programming, and enabling us to simulate a DRNN for speech recognition with an order of magnitude higher parameters than digit recognition networks (Fig. 4d). Thus, our work extends the frontiers of current neuromorphic devices by enabling unprecedented accuracy and scale of parameters required for online, adaptive and truly intelligent systems for applications in speech recognition and natural language processing. Modulating the generation and recombination dynamics of photocarriers, the proposed optoelectronic architecture can be extended to a wide variety of photoresponsive semiconducting platforms (Supplementary Note 9, Supplementary Figs. 22–25). By employing semiconductors of varying band-gap and optical absorption, this concept can utilise wavelength division multiplexing schemes, enabling selective probing of artificial neural networks. In comparison to previous works that demonstrate optical programming and electrical erase[50–52], a $V_{gs}$ that keeps the trap states empty, a $V_{ds}$ that ensures the leakage currents do not interfere with the weight readouts, a low initial background carrier concentration to prevent screening effects and a light intensity sufficient to fill the traps ensures the better linear update of weights in our devices (Supplementary Note 7, Supplementary Figs. 12–16 and

Supplementary Note 9, Supplementary Figs. 22–25). Our concepts could be realised in larger scales by leveraging on the developments of 2D TMDC wafer scale device arrays and heterointegration concepts[53–55]. In short, our optoelectronic neuromorphic computing platform would allow memristive-based implementations to advance beyond simple pattern matching to complex cognitive tasks such as visual question answering, machine translation and dialogue generation.

## Methods

**Device fabrication.** A scotch-tape method was used to exfoliate $ReS_2$ flakes from bulk crystal and was transferred onto a degenerately doped Si substrate with 285 nm $SiO_2$. The electrodes were patterned via photolithography, followed by thermal evaporation of Cr/Au (5/50 nm) and subsequent lift-off process. Height profiling with atomic force microscopy (AFM) was conducted to determine the thickness of the sample. Raman characterisation was performed to confirm the purity of the sample.

**I–V measurements.** For all the optoelectronic characterisations in the manuscript, the PENs were subjected to input optical pulse trains of constant pulse width and interval using an LED light source (ThorLabs SOLIS-445C) equipped with a DC2200 driver. The device conductance was measured using a parameter analyser (Keithley 4200SCS) in sync with the LED light source.

## Data availability

The data that support the findings of this study are available from the corresponding author upon reasonable request.

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

## Acknowledgements
We would like to acknowledge the funding from MOE Tier 1 grants: RG87/16, RG 166/16 and MOE Tier 2 grants: MOE2015-T2-2-007, MOE2015-T2-2-043, MOE2017-T2-2-136 and MOE Tier 2 grant MOE2016-T2-1-100.

## Author contributions
R.A.J., A.B., and N.M. conceived the experiments. R.A.J. performed all the optoelectronic characterizations under the supervision of N.M.; J.A. performed all the neural network simulations and S.K.B. designed the LIF neuron circuit under the supervision of A.B.; C.Z., A.C., Y.G., Y.H., K.K.Z., and X.M. fabricated the devices under the supervision of Z.L.; A.S./N.T. fabricated the P3HT/IWO memtransistors under the supervision of W.L.L./N.M.; R.A.J., A.B., and N.M. wrote the manuscript with comments from all authors.

## Competing interests
The authors declare no competing interests.
