## [Peer Review File · Nature Communications]

Reviewers' comments:

Reviewer #1 (Remarks to the Author):

The manuscript by John et al. reported a 2D FET with optical modulation of the channel conductance, which simulated neural integrate-and-fire and synaptic plasticity. The linear potentiation and depression benefited deep recurrent networks for speech classification. The results are interesting and timely. The following concerns shall be addressed:

(1) It may be less precise to say limited weights can only accommodate shallow networks. (arXiv:1609.07061) Quantized weights indeed reduce the possible sampling points of the space of trainable parameters, so the optimizer is less likely to identify the global minimum of the loss function. However, this could usually be compensated by increasing the population of the weights.

(2) The channel conductance change in Fig. 1B seems to be volatile, which faded after the cease of the illumination. However, the STDP, illustrated in Fig. 1C, seems to be non-volatile? Could the authors briefly discuss what's the mechanism that the illumination could permanently impact the channel conductance, when it's paired with the pre- and post- electrical pulses.

(3) Fig. S5, the right most panel. It seems that the conductance change is non-volatile, as it didn't decay around 275s. In that case, why a fire-threshold is defined?

(4) Fig. 2E-J may be better visualized with statistical box plots.

(5) For abacus demonstration, it seems that the PENs in Fig. 2K were excited by different wavelengths?

(6) Could the authors comment what's the challenge in training the DRNN with ADAM? It has been reported that RMSprop can cope with non-precise weights in online training relatively simple CNN or ConvLSTM. (DOI:10.1038/s42256-019-0089-1)

Reviewer #2 (Remarks to the Author):

In the manuscript titled "Optogenetics-Inspired Light-Driven 2D TMDC Computational Circuits Enable In-Memory Computing for Deep Recurrent Neural Networks", the authors investigated a unique neuromorphic computing platform comprising of optoelectronically-modulated artificial neurons and synapses with selective excitability and linear conductance changes, sufficing blind weight-update protocols for highly-parallel DRNNs. In addition, in-memory computations across 980 addressable states with an average signal-to-noise ratio as high as 77 utilizing photo-excitabile neuristors was demonstrated and this extremely high linear dynamic range was exploited to simulate a memristive DRNN to recognize spoken commands with >90% accuracy. The reviewer thinks that researchers in this field would be very interested in the contents of this manuscript. Therefore, the reviewer would recommend the publication of this paper to Nature Communications after the following minor revision.

- Please add references related with the sentence "speech recognition and natural language processing (NLP) require learning of temporal signals, recurrent connections and deeper architectures (> 10 hidden layers) as in DRNNs." in line 24-27, page 2.

- The author said that this system was inspired from optogenetics and the reviewer thinks that the light intensity (e.g., 230 mW/cm²) used in this study was too high for actual applications. Is this light intensity comparable to optogenetics?

- The author's approach is to selectively activate either neurons or synapses in an artificial neural

network. But, in Fig. 1D, it seemed that specific region was intentionally blocked for light exposure not to be activated for light exposure. That is, the PEN did not show selective activation for light exposure.

- The author said that this modulation of STDP windows resemble biological optogenetic measurements in line 17-20, page 5. Then, in optogenetic measurements, are the shift of STDP functions observed? Please explain in detail and add the related references for abroad ranges of readers.

- The author mentioned programming weights with optical pulses resulted in near linear changes of the STDP function with dosage of incident light in line 1, page 6 and this study demonstrated such linear weight updates with light stimuli in line 15 page 8. Then, does the author think the reason of linear programming is optical modulation? However, photo-generated electron-hole pairs are not monotonically increased with the dosage of incident light. Even, they are saturated at some point. The mechanism of linear change of conductivity should be dealt with at the device level in the manuscript.

- In addition, what is the mechanism of linear decrease under electronic erasing process? Usually, other electrical erasing processes (application of voltage) have not shown this extremely linear modulation of conductivity [Adv. Funct. Mater. 2018, 28, 1804397; Nat. Commun. 2018, 9, 5106; Adv. Funct. Mater. 2019, 29, 1902374]

- Was Figure 2 obtained by using only PEN? Or I&F circuit in Fig 1B? This was ambiguously stated in the manuscript. In addition, this study stated only about device characteristics. To satisfy the high standard of Nat. Commun. Journal, the operating principle of PEN should be demonstrated in the manuscript.

- Please explain the equipment setting and measurement process for making light pulses (e.g., inset in Fig. 2A) as an experimental section.

- Font size is too small in all figures. Please change them.

- The reviewer recommends the authors to include the following references related with neuromorphic hardware solutions in the manuscript.

Adv. Mater. 2019, 1903558

Nat. Nanotechnol. 2019, 14, 776

Reviewer #3 (Remarks to the Author):

This paper shows in-memory computing for DRNN using photoactive ReS₂ based circuit. They also claim resemblance with optogenetics. After going through the manuscript I am not sure what is really novel at the device, circuit or algorithm level that will make this paper appealing for a broad readership of Nature Communications. The term optogenetics appears to be simply a selling point for the paper. Examples of optical write and electrical read or vice versa or both can be found abundantly in the literature both using 2D materials and others. Similarly, in a synaptic cross-bar array one can access each individual node through select lines (like word line and bit line in a content addressable memory). Relating these to optogenetics is rather naive. Linear weight update, STDP, wavelength dependence of photoconductance are all well known. Neither any new physics nor any new material property is demonstrated. Furthermore, each neuron uses a complex circuit for spike generation which increases the synaptic footprint and complicates architectural implementation. In short the PEN defies energy, size and complexity scaling requirements for non von Neumann Computing. I do not recommend publication of this manuscript in Nature Communications

Reviewer #1 (Remarks to the Author):

The manuscript by John et al. reported a 2D FET with optical modulation of the channel conductance, which simulated neural integrate-and-fire and synaptic plasticity. The linear potentiation and depression benefited deep recurrent networks for speech classification. The results are interesting and timely. The following concerns shall be addressed:

We thank the referee for the positive comments on the significance of our work. Our responses to the comments are as follows.

(1) It may be less precise to say limited weights can only accommodate shallow networks. (arXiv:1609.07061) Quantized weights indeed reduce the possible sampling points of the space of trainable parameters, so the optimizer is less likely to identify the global minimum of the loss function. However, this could usually be compensated by increasing the population of the weights.

We thank the reviewer for this comment. Indeed, increasing the population of weights can help in compensating the accuracy—but this again increases the memory and computations required by the network thus partially offsetting the gains achieved by quantization. What we meant by limited precision weights being suitable for shallow architectures is observable from multiple references (2018. *Proceedings of the European conference on computer vision (ECCV)* (pp. 365-382), 2016, *European conference on computer vision* (pp. 525-542). Springer, Cham.,) and is also linked to the dataset complexity. The drop in accuracy after quantization is minimal for relatively shallow networks used to classify simpler datasets such as MNIST, SVHN and CIFAR-10 (<1% drop). However, for deeper networks used to classify more complicated datasets like Imagenet, the accuracy loss due to quantization is much more (~10% or more) (arXiv:1609.07061). Similarly, the required precision for weights in recurrent networks such as RNN or LSTM are higher than those of the shallow, feed-forward networks in this reference (2018. *Frontiers in Neuroscience*, 12, p.745.).

We have now slightly modified the earlier sentence in our introduction to read: “Such limited precision weights can only be used for shallow feed-forward networks trained to classify simple datasets and are incapable of addressing applications like speech recognition and natural language processing (NLP) that require learning of temporal signals, recurrent connections and deeper architectures (>10 hidden layers) as in DRNNs.” (Please refer to the Introduction section Pages 2-3). We have also modified our abstract (Lines 1-5) now to indicate significance of DRNNs and their higher bit precision requirements more clearly.

(2) The channel conductance change in Fig. 1B seems to be volatile, which faded after the cease of the illumination. However, the STDP, illustrated in Fig. 1C, seems to be non-volatile? Could the authors briefly discuss what's the mechanism that the illumination could permanently impact the channel conductance, when it's paired with the pre- and post-electrical pulses.

Yes the reviewer is correct that weight changes depicted in Fig. 1B is volatile while those depicted in Fig. 1C is non-volatile. This is due to the difference in the input programming light pulses used for both measurements. In Figure 1B, the input light (wavelength=445nm) programming pulse used had intensity=5mW/cm², pulse width=10ms while for Figure 1C the corresponding values were intensity=230mW/cm², pulse width=60secs. This is now explicitly mentioned this in the figure captions as well as the legends of Figures 1B-C.

Mechanism for permanent change in the channel conductance: Upon photoexcitation, there is an increase in the carrier densities which reduce through recombination for short illumination pulse widths observed as a short photocurrent decay (as noted in Fig 1B). However for long illumination pulse widths, the photocurrents are maintained even after ceasing the illumination. This long lived change in channel conductance in our FETs is due to a phenomenon called persistent photoconductivity (PPC). Materials with PPC experience an increase in conductivity upon exposure to light that persists after the light is turned off. This is essentially a result of delayed recombination of the generated photocarriers due to their localisation in traps or defects. It has been observed in a wide range of materials including:

(i) Organic semiconductors (2015. *Advanced Materials*, 27(2), pp.228-233., 2016. *Advanced Materials*, 28(16), pp.3078-3086., 2016. *ACS Energy Letters*, 1(5), pp.906-912.)

(ii) Metal oxide semiconductors (2012. *Nature Materials*, 11(4), pp.301-305., 2015. *Science Advances*, 1(9), p.e1500640.)

(iii) III-V semiconductors (1997. *Applied Physics Letters*, 71(8), pp.1098-1100.)

(iv) 2D transition metal dichalcogenide (TMDC) semiconductors (2013. *Nature Nanotechnology*, 8(7), pp.497-501., 2018. *ACS Applied Materials & Interfaces*, 10(42), pp.36512-36522).

However, the exact nature of these traps or defects vary with these systems and is still an active topic of research (2019. *Advanced Functional Materials*, 29(45), p.1905657., 2019. *IEEE Journal of Quantum Electronics*, 56(1), pp.1-5., 2020. *arXiv:2001.04690*).

In TMDCs, earlier studies have indicated different possible origins for the electron trapping and detrapping mechanisms.

1. *Surface Adsorbates*- Gas molecules (e.g., oxygen and water) from the environment are easy to be adsorbed on the surface of TMDCs, leading to large trapping probabilities in FETs. This manifests in reduced conductivities and threshold voltage instabilities in operation as observed in a number of investigations. (Rao, C.N.R., et al. 2012. *ACS Nano*, 6(6), pp.5635-5641., Li et al. 2014. *Applied Physics Letters*, 105(9), p.093107. and Cho et al. 2013. *ACS Nano*, 7(9), pp.7751-7758.

2. *Electron trapping at the semiconductor-dielectric interface*- Interface trap states are considered to originate from the dangling Si–O bonds at the surface. Carriers trapped in these interface states follow a Fermi–Dirac distribution, with the Fermi level being determined by the equilibrium Fermi level of the MoS₂ channel. Guo et al. (2015. *Applied Physics Letters*, 106(10), p.103109.) and Park et al. (2016. *Applied Physics Letters*, 108(8), p.083102.).

3. *Intrinsic semiconductor defects*- Charge carrier trapping has also been observed in suspended MoS₂ FETs measured under high-vacuum conditions. In such devices the origin of traps was attributed to the defects in the TMDC itself. Shu et al., (2016. *Nanoscale*, 8(5), pp.3049-3056., 2017. *npj 2D Materials and Applications*, 1(1), pp.1-9.)

Since our measurements were performed in high-vacuum conditions, the probability of surface adsorbates are highly reduced. Thus, we attribute the origin of trapping-detrapping mechanism in our measurements to defects in the semiconductor itself and the at the semiconductor-

dielectric interface. The high value of subthreshold slope ($\sim 2.5\text{V/dec}$) in our devices support the presence of traps influencing the ReS₂ channel in accordance with similar measurements in 2013. *Nature Nanotechnology*, 8(7), pp.497-501. and 2018. *ACS Applied Materials & Interfaces*, 10(42), pp.36512-36522. In the present work, we utilize a combination of optical and electrical pulses to fill and empty these traps and hence, induce non-volatile conductance changes in our FETs.

Hence, we believe that during each optical “write” operation, the increase in conductance corresponds to the photo-generation of carriers in the semiconducting channel and their delayed recombination, which results in a conductance state that remains stable -demonstrating excellent non-volatility and retention due to the persistent photoconductivity effect. In n-type semiconductors (such as ReS₂ in our case), holes are expected to be trapped while the reverse is expected for p-type semiconductors (2017 *Adv. Sci.* 4, 1700323; *Nanotechnology* 28 (2017) 214002; 2013. *Nature Nanotechnology*, 8(7), pp.497-501; 2012. *Nature Materials*, 11(4), pp.301-305.) These trapped minority carriers have also been referred to as the cause of a “photogating” effect. The subsequent photo-generation pulses adds on to the carrier concentration resulting in a near-ideal conductance linearity. The number of distinct states are determined primarily by the programming pulse resolution and recombination kinetics of the photo-generated carriers, and hence, the programming pulses could be chosen to achieve a near-perfect linearity.

While optical pulses enables linear incremental non-volatile “write” steps, electrical gating is used to “erase” via defect-assisted recombination. Here, the “erase” process modulated via the electrical gating raises the Fermi level to induce electron accumulation in the channel and accelerates the recombination of electrons with the holes in the trap sites resulting in lowered current levels as shown in Figures-2B-D and Supporting Information Figure-S9. This is in line with the conclusions of Robertson, J., et al. in 2012. *Nature Materials*, 11(4), pp.301-305. and 2013. *Nature Nanotechnology*, 8(7), pp.497-501 where voltage pulses that induced majority carriers were used to eliminate persistent photocurrent. We have now included this discussion in the main text (Please refer to Page 8).

(3) *Fig. S5, the right most panel. It seems that the conductance change is non-volatile, as it didn't decay around 275s. In that case, why a fire-threshold is defined?*

We demarcated a firing threshold in Figure S5 just to highlight the fact that the difference in photoresponse or wavelength selectivity of our artificial synapses would allow a wide flexibility for setting the threshold for neuronal firing. In other words, we foresee development of artificial synapses that respond to narrow bands of optical pulses in future and would like to point out that our approach would enable setting of desired pre-set thresholds for selective neuronal firing when such artificial synapses are connected to artificial neurons. We have now amended the and supporting information (Figure S5) to denote this.

(4) *Fig. 2E-J may be better visualized with statistical box plots.*

We thank the reviewer for his/her comment. As per the reviewer’s suggestion, we have now replotted Figures 2E-J as box and whisker plots. Please refer to the new Figure 2 in the main text (also shown below).

(5) For abacus demonstration, it seems that the PENs in Fig. 2K were excited by different wavelengths?

No. We would like to clarify that the optical pulses ($\lambda=445,525$ and 623nm , pulse width= 10s , intensity= $65\text{mW}/\text{cm}^2$) used for programming the PENs were the same throughout the manuscript. For Figure 2K, the bead colours correspond to the illumination wavelength used. The step size of conductance updates shown in Figure 2K are exactly same as those shown in Figures 2A-D, SI-Note-4 Figures S6-S8. Example: red bead corresponding to $\lambda=623\text{nm}$ has a step size of 0.15nS in Figure 2K, similar to the ones depicted in Figure S6 ($0.15\text{-}0.18\text{nS}$).

(6) Could the authors comment what's the challenge in training the DRNN with ADAM? It has been reported that RMSprop can cope with non-precise weights in online training relatively simple CNN or ConvLSTM. (DOI:10.1038/s42256-019-0089-1)

Thank you for the comment. The reference pointed out by the reviewer does online training in a limited sense—they measure the memristive conductances, and then calculated the required change in conductance offline using a digital computer and finally write those conductance changes back to the memristor array. While this work shows that the nonlinearities in the blind update can be tackled, it does not really provide a solution to the gradient calculation which is still done offline on a digital computer. In that sense, there is very limited acceleration provided to the training process where each weight is read and updated multiple times. In contrast, we perform the entire training offline and then write the weights using our two-shot write scheme. We can do this due to our extremely high linearity and SNR of weight update. Fully parallel weight update schemes (2018. *Nature Communications*, 9(1), pp.1-8., 2015. *IEEE TNNLS* 26(10), pp.2408-2421.) can accelerate the gradient based update much better since the weight update is performed in one shot for one crossbar. However, such schemes can only implement stochastic gradient descent (SGD) without momentum and produce inferior results compared to using ADAM optimizer. Hence, we believe our solution of offline training using sophisticated optimizers like ADAM followed by two-shot write based weight transfer provides a promising solution to implement energy efficient neuromorphic DRNN. We have now amended the main text at several places in the main text to reiterate that offline training with ADAM is what we have presented in this work (Please refer to Pages 4 and 12). Details of ADAM and the novel Two-shot Write Scheme are now provided in the supporting information (Please refer to Note-7 Pages 17-19).

Reviewer #2 (Remarks to the Author):

In the manuscript titled “Optogenetics-Inspired Light-Driven 2D TMDC Computational Circuits Enable In-Memory Computing for Deep Recurrent Neural Networks”, the authors investigated a unique neuromorphic computing platform comprising of optoelectronically-modulated artificial neurons and synapses with selective excitability and linear conductance changes, sufficing blind weight-update protocols for highly-parallel DRNNs. In addition, in-memory computations across 980 addressable states with an average signal-to-noise ratio as high as 77 utilizing photo-excitabile neuristors was demonstrated and this extremely high linear dynamic range was exploited to simulate a memristive DRNN to recognize spoken commands with >90% accuracy. The reviewer thinks that researchers in this field would be very interested in the contents of this manuscript. Therefore, the reviewer would recommend the publication of this paper to Nature Communications after the following minor revision.

We thank the reviewer for commending our contribution and pointing out the significance of our experimental demonstration of linear weight updates with optoelectronically-modulated artificial synapses. We also very much appreciate the reviewer’s suggestions to improve our manuscript by making more objective statements.

- Please add references related with the sentence “speech recognition and natural language processing (NLP) require learning of temporal signals, recurrent connections and deeper architectures (> 10 hidden layers) as in DRNNs.” in line 24-27, page 2.

We thank the reviewer for this comment. We have now added the following references to validate this sentence.

1. Park, Daniel S., et al. "SpecAugment: A simple data augmentation method for automatic speech recognition." arXiv preprint arXiv:1904.08779 (2019).
2. Devlin, Jacob, et al. "Bert: Pre-training of deep bidirectional transformers for language understanding." arXiv preprint arXiv:1810.04805 (2018).

- The author said that this system was inspired from optogenetics and the reviewer thinks that the light intensity (e.g., 230 mW/cm²) used in this study was too high for actual applications. Is this light intensity comparable to optogenetics?

We thank the reviewer for this comment. In biological optogenetics measurements, the light pulses used vary across intensities of 100-10,000mW/cm² depending on the type of neuronal cells targeted and the photosensitive proteins employed [2015. *Nat. Neurosci*, 18(9), pp.1213-1225., 2013. *Nat. Neurosci*, 16(7), p.805., 2011. *Annual Review of Neuroscience*, 34., 2019. *Frontiers in Physiology*, 10, p.1096.]. This corresponds well with the light source intensity we utilize for this work (65-230mW/cm²).

- The author’s approach is to selectively activate either neurons or synapses in an artificial neural network. But, in Fig. 1D, it seemed that specific region was intentionally blocked for light exposure not to be activated for light exposure. That is, the PEN did not show selective activation for light exposure.

We thank the reviewer for this comment. We would like to clarify that in this experiment the light was intentionally blocked to rule out possibilities of even a minuscule amount of stray photocurrent in the unselected synapses. However, we understand the reviewer’s concern regarding the demonstration of selectivity and hence to address this, we have now conducted additional experiments without blocking light intentionally. To illustrate the selective activation further, a new 36-element array comprising of 16 light sensitive ReS₂ and 20 insensitive Indium Tungsten Oxide (IWO) synapses was assembled as shown in Figure-1E. All

synapses were initially programmed to a common low conductance state (5nS). Upon illumination with red light ($\lambda=623$ nm, 65mW/cm² intensity), the channel conductance of the ReS₂ PENs depicted a non-volatile increment (5 to 6.5nS) due to its low bandgap (1.5–1.8eV). On the other hand, the channel conductance of IWO synapses remained constant (5nS) due to their large bandgap (3.6eV). A spatial conductance map readout of the array resembled the image of “N” with the light sensitive ReS₂ synapses spatially arranged to resemble this alphabet as shown in Figure-1E. These results demonstrate the specificity of a light-based “write” operation, globally capable of addressing light-sensitive neural elements on demand. **The new results are presented as Figure 1E (also shown below).**

- *The author said that this modulation of STDP windows resemble biological optogenetic measurements in line 17-20, page 5. Then, in optogenetic measurements, are the shift of STDP functions observed? Please explain in detail and add the related references for abroad ranges of readers.*

We thank the reviewer for this comment. Traditionally in biology, optogenetics was used to activate a selected patch of neurons[*Nat. Methods* **11**, 1012 (2014).]. Recently, this technique has advanced to probe synapses and other cortical structures[2015. *Nat. Neurosci*, **18**(9), pp.1213-1225., 2013.]. **We have briefly explained about optogenetics and its advances in the introduction (Please refer to Page 3 in the main text).** Shift in the STDP functions have been observed in biological measurements [Eg: 2018. *Neuron*, **98**(4), pp.801-816., 2019. *bioRxiv*, p.863365., 2018., *Neuron*, **97**(6), pp.1244-1252., 2015. *Neuron*, **88**(3), pp.528-538.], similar to our observations. **As per the reviewer’s suggestion, we have now included some of the above references in the main text.**

- *The author mentioned programming weights with optical pulses resulted in near linear changes of the STDP function with dosage of incident light in line 1, page 6 and this study demonstrated such linear weight updates with light stimuli in line 15 page 8. Then, does the author think the reason of linear programming is optical modulation? However, photo-generated electron-hole pairs are not monotonically increased with the dosage of incident light. Even, they are saturated at some point. The mechanism of linear change of conductivity should be dealt with at the device level in the manuscript.*

In addition, what is the mechanism of linear decrease under electronic erasing process? Usually, other electrical erasing processes (application of voltage) have not shown this extremely linear modulation of conductivity [Adv. Funct. Mater. 2018, 28, 1804397; Nat. Commun. 2018, 9, 5106; Adv. Funct. Mater. 2019, 29, 1902374]

We thank the reviewer for these 2 comments. Yes we believe that optical modulation results in a linear programming of weights.

Mechanism for permanent change in the channel conductance: Upon photoexcitation, there is an increase in the carrier densities which reduce through recombination for short illumination pulse widths observed as a short photocurrent decay (as noted in Fig 1B). However for long illumination pulse widths, the photocurrents are maintained even after ceasing the

illumination. This long lived change in channel conductance in our FETs is due to a phenomenon called persistent photoconductivity (PPC). Materials with PPC experience an increase in conductivity upon exposure to light that persists after the light is turned off. This is essentially a result of delayed recombination of the generated photocarriers due to their localisation in traps or defects. It has been observed in a wide range of materials including:

(i) Organic semiconductors (2015. *Advanced Materials*, 27(2), pp.228-233., 2016. *Advanced Materials*, 28(16), pp.3078-3086., 2016. *ACS Energy Letters*, 1(5), pp.906-912.)

(ii) Metal oxide semiconductors (2012. *Nature Materials*, 11(4), pp.301-305., 2015. *Science Advances*, 1(9), p.e1500640.)

(iii) III-V semiconductors (1997. *Applied Physics Letters*, 71(8), pp.1098-1100.)

(iv) 2D transition metal dichalcogenide (TMDC) semiconductors (2013. *Nature Nanotechnology*, 8(7), pp.497-501., 2018. *ACS Applied Materials & Interfaces*, 10(42), pp.36512-36522).

However, the exact nature of these traps or defects vary with these systems and is still an active topic of research (2019. *Advanced Functional Materials*, 29(45), p.1905657., 2019. *IEEE Journal of Quantum Electronics*, 56(1), pp.1-5., 2020. *arXiv:2001.04690*.)

In TMDCs, earlier studies have indicated different possible origins for the electron trapping and detrapping mechanisms.

1. *Surface Adsorbates*- Gas molecules (e.g., oxygen and water) from the environment are easy to be adsorbed on the surface of TMDCs, leading to large trapping probabilities in FETs. This manifests in reduced conductivities and threshold voltage instabilities in operation as observed in a number of investigations. (Rao, C.N.R., et al. 2012. *ACS Nano*, 6(6), pp.5635-5641., Li et al. 2014. *Applied Physics Letters*, 105(9), p.093107. and Cho et al. 2013. *ACS Nano*, 7(9), pp.7751-7758.

2. *Electron trapping at the semiconductor-dielectric interface*- Interface trap states are considered to originate from the dangling Si–O bonds at the surface. Carriers trapped in these interface states follow a Fermi–Dirac distribution, with the Fermi level being determined by the equilibrium Fermi level of the MoS₂ channel. Guo et al. (2015. *Applied Physics Letters*, 106(10), p.103109.) and Park et al. (2016. *Applied Physics Letters*, 108(8), p.083102.).

3. *Intrinsic semiconductor defects*- Charge carrier trapping has also been observed in suspended MoS₂ FETs measured under high-vacuum conditions. In such devices the origin of traps was attributed to the defects in the TDMC itself. Shu et al., (2016. *Nanoscale*, 8(5), pp.3049-3056., 2017. *npj 2D Materials and Applications*, 1(1), pp.1-9.).

Since our measurements were performed in high-vacuum conditions, the probability of surface adsorbates are highly reduced. Thus, we attribute the origin of trapping-detrapping mechanism in our measurements to defects in the semiconductor itself and the at the semiconductor-dielectric interface. The high value of subthreshold slope (~2.5V/dec) in our devices support the presence of traps influencing the ReS₂ channel in accordance with similar measurements in 2013. *Nature Nanotechnology*, 8(7), pp.497-501. and 2018. *ACS Applied Materials & Interfaces*, 10(42), pp.36512-36522. In the present work, we utilize a combination of optical

and electrical pulses to fill and empty these traps and hence, induce non-volatile conductance changes in our FETs.

Hence, we believe that during each optical “write” operation, the increase in conductance corresponds to the photo-generation of carriers in the semiconducting channel and their delayed recombination, which results in a conductance state that remains stable -demonstrating excellent non-volatility and retention due to the persistent photoconductivity effect. In n-type semiconductors (such as ReS₂ in our case), holes are expected to be trapped while the reverse is expected for p-type semiconductors (2017 *Adv. Sci.* 4, 1700323; *Nanotechnology* 28 (2017) 214002; 2013. *Nature Nanotechnology*, 8(7), pp.497-501; 2012. *Nature Materials*, 11(4), pp.301-305.) These trapped minority carriers have also been referred to as the cause of a “photogating” effect. The subsequent photo-generation pulses adds on to the carrier concentration resulting in a near-ideal conductance linearity. The number of distinct states are determined primarily by the programming pulse resolution and recombination kinetics of the photo-generated carriers, and hence, the programming pulses could be chosen to achieve a near-perfect linearity.

Mechanism for linear conductance changes: These investigations (2013. *Nature Nanotechnology*, 8(7), pp.497-501. and 2018. *ACS Applied Materials & Interfaces*, 10(42), pp.36512-36522.) also demonstrate the possibility of tuning the photoconductivity across a wide range by appropriate choice of gate and drain voltages, in turn switching the operation from a photogating (PG) to a photoconductive (PC) regime (and vice versa) by modulating trap occupancies. From similar investigations, it is clear that biasing the FETs to their off condition ($V_G < V_{TH}$) would ensure the bulk and interface traps are mostly empty owing to a lack of carriers in the channel. This would allow a large number of photogenerated charge carriers to get trapped leading to a prominent photogating effect (slow and large increase in current) and persistent photoconductivity visible in the photocurrent dynamic response. This is in line with our biasing conditions as well. We bias all our devices to their off state before starting our measurements as indicated in SI-Note-3 Figure-S5. This maximizes the available linear/triode region of the FET to probe maximum number of states in a linear manner using optical pulses. This negative bias is also reported to cause a physical separation of photogenerated electron-hole pairs by confining the electrons to the back channel, even after the illumination has stopped, compounding the PPC effect (2012. *Nature Materials*, 11(4), pp.301-305), again in alignment with our measurements.

While optical pulses enables linear incremental non-volatile “write” steps, electrical gating is used to “erase” via defect-assisted recombination. Here, the “erase” process modulated via the electrical gating raises the Fermi level to induce electron accumulation in the channel and accelerates the recombination of electrons with the holes in the trap sites resulting in lowered current levels as shown in Figures-2B-D and Supporting Information Figure-S9. This is in line with the conclusions of Robertson, J., et al. in 2012. *Nature Materials*, 11(4), pp.301-305. and 2013. *Nature Nanotechnology*, 8(7), pp.497-501 where voltage pulses that induced majority carriers were used to eliminate persistent photocurrent. With regards to the linear nature of erase, we would like to clarify that this is made possible by carefully selecting the erase voltages as shown in the Figure below (This figure is now added to the supporting information Figure-S9F). Increasing the gate voltage allows for accumulating additional electrons which can recombine with the trapped holes, allowing us to tune our symmetry. This is similar to strategies used by other groups to control conductance levels [2018. *Science Advances*, 4(4), p.eaap7916 and 2015. *Science Advances*, 1(9), p.e1500640]. We have now included this discussion in the main text (Please refer to Page 8).

Saturation of photo-generated carriers: The reviewer is correct that there is a final saturation of these photo-generated carriers. To test the limits of our device, additional experiments were conducted and results are plotted below. We observe that the linear weight updates continue up to ~1018 optical writing pulses for blue light illumination (65mW/cm²) after which the update step size gradually ceases. We clarify that these limits may vary from device to device. The value reported in the manuscript (980 addressable states) is an average and the range of the number of addressable states experimentally measured from our devices is ~900-1053. Moreover, from an algorithm perspective, the exact saturation level is not important as long as the dynamic range is high enough. The 2-shot writing scheme we adopt takes into account the more critical variation in the slope of the curve to eliminate any sort of mismatches at the network level. We have now included this data in the SI-Note-5 Figure S9E.

In summary, the exquisite write linearity afforded by the optical gating is the major phenomenon we exploit in this work to get very accurate weights in a DRNN.

- Was Figure 2 obtained by using only PEN? Or I&F circuit in Fig 1B? This was ambiguously stated in the manuscript. In addition, this study stated only about device characteristics. To satisfy the high standard of Nat. Commun. Journal, the operating principle of PEN should be demonstrated in the manuscript.

- Please explain the equipment setting and measurement process for making light pulses (e.g., inset in Fig. 2A) as an experimental section.

We thank the reviewer for these 2 comments. Figure 2 was obtained by using only PEN. The I&F circuit in Figure 1B was not used for Figure 2. For Figure 2, the PENs were subjected to an input optical pulse train of constant pulse width and interval using an LED light source (ThorLabs SOLIS-445C) equipped with a DC2200 driver. The device conductance was measured using a parameter analyser (Keithley 4200SCS) in sync with the LED light source. We have now added an experimental section to the main text describing the device fabrication, physical and electrical characterizations conducted in the manuscript. We apologize for not having included this in the earlier version of the manuscript. Please refer to the above answer for the operating principle of linear weight updates in PENs.

- Font size is too small in all figures. Please change them.

We have increased the font sizes in several figures and will also consult with the editorial office of the journal to see what best can be done. We thank the reviewer for his/her comment.

- The reviewer recommends the authors to include the following references related with neuromorphic hardware solutions in the manuscript.

Adv. Mater. 2019, 1903558

Nat. Nanotechnol. 2019, 14, 776

As per the reviewer's suggestion, we have now added the above 2 references to the manuscript.

Reviewer #3 (Remarks to the Author):

This paper shows in-memory computing for DRNN using photoactive ReS2 based circuit. They also claim resemblance with optogenetics. After going through the manuscript I am not sure what is really novel at the device, circuit or algorithm level that will make this paper appealing for a broad readership of Nature Communications. In short the PEN defies energy, size and complexity scaling requirements for non von Neumann Computing. I do not recommend publication of this manuscript in Nature Communications.

We thank the reviewer for his/her critical and valuable comments. We have modified the manuscript to better highlight the uniqueness and advantages of our approach. We would like to point out the following novel features of our work to the reviewer. **Most of the published works target shallow feed forward networks** (Eg: 2018, Nature Communications, 9(1), pp.1-8.). While very recent works like 2019, Science, 364(6440), pp.570-574 and 2017, Nature Materials, 16(4), pp.414-418 target neural accelerators that demand excellent linearity, **none of the published works deal with really deep networks** like the 10-layer deep recurrent neural networks (DRNNs) we target in this work. Moreover, **none of the published works portray devices with excellent linearity and high linear dynamic range** required to address the switching issues of DRNNs **necessary for applications such as natural language processing and speech recognition**. The exquisite write linearity afforded by the optical gating is the major phenomenon we exploit to get very accurate weights in a DRNN. We have now modified the abstract to highlight what we target, the significance of our work and the approach we take. We have also now inserted sub-headings within the main text to improve the readability of the manuscript. This work addresses these issues by adopting a novel optoelectronic programming scheme and portrays the following key points:

Novelty at the device/material level:

1. **The reported number of accessible non-volatile states (980 or ~10-bit equivalent) is the highest till date to the best of our knowledge. The SNR values as high as 77 are also among the highest reported till date.** Our devices are superior to pure-electrical and optical memristors in terms of conductance linearity and write noise as detailed in Figure 2 (main text) and Figures-S6-9 (supporting information).

2. We report the first prototypical elements of an optoelectronic neuromorphic computing platform with neuron excitability and synaptic weight updates controlled synergistically via optical and electrical pulses. We demonstrate the selective ability to address neurons, a hallmark of optogenetics, hitherto not demonstrated for hardware neuromorphic circuits.

3. In summary, the concept of utilizing blind linear photo-updates to manipulate the learning behaviours of artificial neural circuitry is the first of its kind. **The proposed optoelectronic architecture can be generalized to a wide variety of semiconducting platforms** (i.e., III-V semiconductors, transition metal dichalcogenides (TMDCs), halide perovskites and organic semiconductors), opening up new possibilities with improved scalability and CMOS compatibility. **To prove the universal nature of the proposed concept, we have carried out additional experiments on Poly(3-hexylthiophene) (P3HT) semiconducting films.** The figure below depicts the linear weight updates in P3HT. Optical pulses enable programming or writing of conductance states, while electrical voltages to the back-gate act as erasing pulses. The source and drain electrodes act as the reading terminals across which the memconductance or memresistance is read. As evident from the graph below, weight updates with excellent linearity can be programmed facilely via optical pulses. By optimising the light intensity and

pulse width, it is possible to access large number of linear non-volatile states.

Pure-electrical 2-terminal memristive implementations have failed to achieve linear weight updates due to their abrupt switching physics [2016. Nature Communications, 7, p.12805., 2018. Nature Communications, 9(1), p.5311.]. 3-terminal memtransistor implementations have also failed in this regard till date [2019. Nature Materials, 18(2), p.141.; 2019. Nature Nanotechnology, 14(8), pp.776-782.; 2018. Nature, 554(7693), p.500]. The novel optoelectronically “gated” memtransistor configuration we adopt allows us to extract an unprecedented number of memory states from our devices by perturbing the charge carrier generation-recombination effects at the semiconductor-dielectric interface via multiple modes.

4. By employing semiconductors of varying band-gap and optical absorption, this concept can be extended to a wide variety of material systems and wavelength division multiplexing schemes, enabling highly selective probing of artificial neural networks. To substantiate selective activation further, a new 36-element array comprising of 16 light sensitive ReS₂ and 20 light insensitive Indium Tungsten Oxide (IWO) synapses was assembled as shown in Figure-1E. All synapses were initially programmed to a common low conductance state (5nS). Upon illumination with red light ($\lambda=623$ nm, 65mW/cm² intensity), the channel conductance of the ReS₂ PENs depicted a non-volatile increment (5 to 6.5nS) due to its low bandgap (1.5–1.8eV). On the other hand, the channel conductance of IWO synapses remained constant (5nS) due to their large bandgap (3.6eV). A spatial conductance map readout of the array resembled the image of “N” with the light sensitive ReS₂ synapses spatially arranged to resemble this alphabet as shown in Figure-1E. These results demonstrate the specificity of a light-based “write” operation, globally capable of addressing light-sensitive neural elements on demand. **The new results are presented as Figure 1E (also shown below).**

Novelty at the circuit/algorithm level:

1. Exploiting the extremely low noise and high linearity in writes of our photo-excitabile neuristors (PENs), we report neural network simulations of vector matrix multiply (VMM) operations within memory. **This is the first report of a memristive device enabled low-**

power end to end speech classification network, which also opens up the option of online learning with complex deep networks for high accuracy. The proposed PEN features an order of magnitude higher linear dynamic range (LDR) than other recent state-of-the-art reports, enabling us to simulate a DRNN for speech recognition with an order of magnitude higher parameters than digit recognition networks. In contrast to other reports in this field which deal with lower number of trainable weights and parameters, we benchmark our devices on large-scale networks to show the superior characteristics of the proposed approach. This again makes it the go-to-report for large-scale neural networks as of today. Hence, we would like to reiterate the significance of the problem we tackle with our devices and the superior features of the proposed approach.

2. Any momentum based learning rule is currently not implementable by the current RRAM based online learning works reported in literature—all of them use stochastic gradient descent [2018. Nature Communications, 9(1), p.2385.; 2017. Nature Nanotechnology, 12(8), p.784.; 2019. Nature Materials, 18(4), p.309]. The momentum based rule we have used is ADAM (D. Kingma and J. Ba, “Adam: A Method for Stochastic Optimization,” arXiv, Dec 2014.) which has been shown to produce state-of-the-art results in many domains.

ADAM has several advantages over traditional stochastic gradient descent. Firstly, it contains a momentum term which accelerates the learning process and prevents it from getting stuck at local optima. Secondly, in ADAM, the learning parameters are adapted on per-parameter basis based on previous values of parameter updates which results in smaller updates for frequent features and larger updates for infrequent features. This improvement is particularly useful for sparse gradients. Since the learning parameter adaptation is done based on only past few parameter updates, this optimization technique is also more useful than SGD for online learning and non-stationary objectives. Thirdly, since the learning parameters in ADAM are adapted on per-parameter basis, it is considered more robust towards choice of hyper parameters. Finally, ADAM calculates the bias-corrected parameter estimations which prevents it from being biased towards initial gradient and momentum values. These advantages have made ADAM the most popular choice for gradient descent optimization in past few years. **We have now**

added more detailed text in the main draft and supporting information to clearly explain the advantages of our approach (Please refer to SI-Note-7).

2. We demonstrate several unique features of optogenetics in our experiments, namely-wavelength and intensity modulations and spatiotemporal selectivity using a prototypical neuromorphic array composed on 36 PENs. **This is the first demonstration of an artificial optoelectronic neuromorphic array with selective activation of neurons and synapses.**

3. Utilizing high-linearity blind writes, we experimentally demonstrate basic arithmetic operations of addition, subtraction, multiplication and division, akin to an abacus, using these non-von Neumann optoelectronic memcomputing devices. **This is the first demonstration of its kind using PENs.**

In short, the proposed concept establishes a new programming scheme for linear blind updates that addresses the switching requirements of deep recurrent neural networks (DRNNs). Existing reports fail to address these issues as described above. Our approach paves way for new optoelectronically “gated” memristive neuromorphic computing architectures.

The term optogenetics appears to be simply a selling point for the paper. Examples of optical write and electrical read or vice versa or both can be found abundantly in the literature both using 2D materials and others. Similarly, in a synaptic cross-bar array one can access each individual node through select lines (like word line and bit line in a content addressable memory). Relating these to optogenetics is rather naive. Linear weight update, STDP, wavelength dependence of photoconductance are all well known. Neither any new physics nor any new material property is demonstrated.

We respectfully disagree with the reviewer on this comment. What we are demonstrating in our manuscript is how even partially applying the principles of optogenetics could bring benefits to the neuromorphic community. The title and the contents of the manuscript clearly states that our approach is only inspired from optogenetics and not a direct correlation to optogenetics (controlling ion channel of biological neurons) like the reviewer has mentioned. We have clearly illustrated the optoelectronic signatures of artificial neurons and synapses via systematic experiments and simulations to prove the utility of the optogenetics inspired approach and their compatibility as fundamental building blocks for artificial neural networks. Traditionally in biology, optogenetics is used to activate a selected patch of neurons[*Nat. Methods* **11**, 1012 (2014).]. Recently, this technique has advanced to probe synapses and other cortical structures[2015. *Nat. Neurosci*, *18*(9), pp.1213-1225., 2013.]. Shift in the STDP functions have been observed in biological measurements [Eg: 2018. *Neuron*, *98*(4), pp.801-816., 2019. *bioRxiv*, p.863365., 2018., *Neuron*, *97*(6), pp.1244-1252., 2015. *Neuron*, *88*(3), pp.528-538.], similar to our observations. We agree with the reviewer that examples of optical write and electrical read do exist in literature. But none of the studies till date have reported comparable linear blind updates necessary for DRNNs. In summary, the exquisite write linearity afforded by the optical gating is the major phenomenon we exploit in this work to get very accurate weights in a DRNN.

Furthermore, each neuron uses a complex circuit for spike generation which increases the synaptic footprint and complicates architectural implementation.

The initial data on light-modulated LIF neuron and STDP-based synapses is presented to benchmark our devices against other spike-based systems using conventional characterizations adopted in the neuromorphic community. We would like to reiterate that the demonstration of neuron circuit and light-modulated STDP synapses was conducted to benchmark our devices against other reports in literature and to demonstrate the correlation from optogenetics. These experiments depict the ease with which weights could be modulated in our devices with optical stimuli, which forms the basis for the subsequent experiments detailed in the manuscript. We do not claim any novelty in the circuit design of the neuron, but in the demonstration of light-modulated firing rate of the neuron, a hallmark of optogenetics.

We have now included additional data in the main text and supporting information, highlighting the novelty of our work. We have now made substantial changes to the manuscript to improve the readability of the novelty and significant findings of our work. A brief overview of the results is now expanded to incorporate more details in the introduction section. Detailed explanation of ADAM and our unique 2-shot write scheme is now provided in the supporting information. Additional experiments have been conducted to prove the selective nature of our optoelectronic programming scheme, universality of this approach to other semiconducting materials and details of the underlying working principle is also now provided for better clarity.

Reviewers' comments:

Reviewer #1 (Remarks to the Author):

In the revision, some of my early questions have been addressed. However, some other concerns still remain. For instance there is a misunderstanding in the answer to my previous comment #6. The authors mentioned that "they measure the memristive conductance, and then calculated the required change in conductance offline using a digital computer and finally write those conductance changes back to the memristor array." This is not accurate since the conductance was not measured for the purpose to implement the weight changes, but merely to do backpropagation. And in principle, the backpropagation can be done in hardware too. The computed gradient is in fact implemented by blind memristor programming using stored gate voltages (similar role with measured conductance) rather than the need to physically measure the conductance. The authors should revise this discussion and update the body text. In addition, a number of important recent progresses in neural networks and in-memory computing demonstrated with emerging devices are missing and should be mentioned in the introduction to give readers a more complete picture of the current status of the field.

Reviewer #2 (Remarks to the Author):

In the manuscript titled "Optogenetics-Inspired Light-Driven 2D TMDC Computational Circuits Enable In-Memory Computing for Deep Recurrent Neural Networks", the authors investigated a unique neuromorphic computing platform comprising of optoelectronically-modulated artificial neurons and synapses with selective excitability and linear conductance changes, sufficing blind weight-update protocols for highly-parallel DRNNs. In addition, in-memory computations across 980 addressable states with an average signal-to-noise ratio as high as 77 utilizing photo-excitabile neuristors was demonstrated and this extremely high linear dynamic range was exploited to simulate a memristive DRNN to recognize spoken commands with >90% accuracy. The reviewer thinks that researchers in this field would be very interested in the contents of this manuscript. Therefore, the reviewer would recommend the publication of this paper to Nature Communications after the following minor revision.

- What is the reason of much higher linearity of programming and erasing weights of the author's device in comparison with other photonic synapses? This property is quite unique, but this was not completely demonstrated. This should be deeply covered in the manuscript.
- Under electronic erasing process, usually, other electrical erasing processes (application of voltage) have not shown this extremely linear modulation of conductivity like the author's device [Adv. Funct. Mater. 2018, 28, 1804397; Nat. Commun. 2018, 9, 5106; Adv. Funct. Mater. 2019, 29, 1902374]. For responding to this question, the author showed Figure S9F, that is, erasing process with various erasing voltage amplitude. But, +30 and +10 V of erasing voltage did not make the device return to the initial points. Even though the previous studies [Adv. Funct. Mater. 2018, 28, 1804397; Nat. Commun. 2018, 9, 5106; Adv. Funct. Mater. 2019, 29, 1902374] completely returned to their initial states with erasing voltages, they did not show linear decrease under electronic erasing process. At the early stage, the conductivity was highly decreased and the amount of decreasing became smaller. Please the reason of higher linearity of the author's device. This is very critical parts in this study.
- Please add the full name of TMDC (title).
- In the response letter, why did the author exclude Intrinsic semiconductor defects as the origin of trapping-detrapping mechanism? Explain this.

Reviewer #3 (Remarks to the Author):

Authors have made some attempts to answer the questions. However, I still think that the manuscript would be better suited for a more specialized journal. I also still do not think that there is any optogenetics motivation here. Showing optical write and electrical erase in one device is easy. But accomplishing the same in a large array is simply impractical.

Reviewer #1 (Remarks to the Author):

1. In the revision, some of my early questions have been addressed. However, some other concerns still remain. For instance there is a misunderstanding in the answer to my previous comment #6. The authors mentioned that "they measure the memristive conductance, and then calculated the required change in conductance offline using a digital computer and finally write those conductance changes back to the memristor array." This is not accurate since the conductance was not measured for the purpose to implement the weight changes, but merely to do backpropagation. And in principle, the backpropagation can be done in hardware too. The computed gradient is in fact implemented by blind memristor programming using stored gate voltages (similar role with measured conductance) rather poses the need to physically measure the conductance. The authors should revise this discussion and update the body text.

We thank the referee for the positive comments on our work and the revision. We agree that the referenced work uses the measured conductance to only backpropagate the error gradient simulating how it would work if the hardware were directly used for backpropagation. However, we would like to clarify that our assertion—weight update with momentum learning is not possible in-situ in memristive crossbar architectures—is still true. We give more details below.

Learning in a hardware neural network can be decomposed into 3 broad phases: (1) error gradient backpropagation, (2) calculation of change in weight Δw based on backpropagated gradient and (3) physical update of weights according to the calculated Δw in step (2). While the authors of 2019. *Nature Machine Intelligence*, 1(9), pp.434-442. could demonstrate that it is possible to accelerate step (1) using a resistive crossbar ideally in $O(1)$ time, steps (2) and (3) can be done in $O(1)$ time in-situ only if the weight update rule is stochastic gradient descent [2016. *Frontiers in neuroscience*, 9, p.484.]. For momentum based learning rules such as RMSprop or ADAM, separate variables have to be maintained for each weight to keep track of its weight update history (Email communication with authors of 2019. *Nature Machine Intelligence*, 1(9), pp.434-442.). In other words, since momentum based weight update cannot be directly written as an outer product of two vectors, it cannot be performed in-situ on a memristive crossbar and requires ex-situ computation for each weight. This justifies our choice of training offline with similar requirement of ex-situ calculation of weights as well. We have now added this discussion in the Supplementary section Note 8, on page 22.

2. In addition, a number of important recent progresses in neural networks and in-memory computing demonstrated with emerging devices are missing and should be mentioned in the introduction to give readers a more complete picture of the current status of the field.

We thank the reviewer for the comment. We have now added additional references throughout the manuscript to depict the recent progresses in neural networks and in-memory computing achieved with emerging devices. The newly added references include:

1. Yao, P. *et al.* Fully hardware-implemented memristor convolutional neural network. *Nature* 577, 641–646 (2020).
2. Wang, Z. *et al.* In situ training of feed-forward and recurrent convolutional memristor networks. *Nat. Mach. Intell.* 1, 434–442 (2019).
3. Agarwal, S. *et al.* Energy scaling advantages of resistive memory crossbar based computation and its application to sparse coding. *Front. Neurosci.* 9, 484 (2016).
4. Wang, Z. *et al.* Resistive switching materials for information processing. *Nat. Rev. Mater.* 1–23 (2020).
5. Prezioso, M. *et al.* Spike-timing-dependent plasticity learning of coincidence detection with passively integrated memristive circuits. *Nat. Commun.* 9, 1–8 (2018).
6. Cai, F. *et al.* A fully integrated reprogrammable memristor–CMOS system for efficient multiply–accumulate operations. *Nat. Electron.* 2, 290–299 (2019).

Reviewer #2 (Remarks to the Author):

In the manuscript titled “Optogenetics-Inspired Light-Driven 2D TMDC Computational Circuits Enable In-Memory Computing for Deep Recurrent Neural Networks”, the authors investigated a unique neuromorphic computing platform comprising of optoelectronically-modulated artificial neurons and synapses with selective excitability and linear conductance changes, sufficing blind weight-update protocols for highly-parallel DRNNs. In addition, in-memory computations across 980 addressable states with an average signal-to-noise ratio as high as 77 utilizing photo-excitabile neuristors was demonstrated and this extremely high linear dynamic range was exploited to simulate a memristive DRNN to recognize spoken commands with >90% accuracy. The reviewer thinks that researchers in this field would be very interested in the contents of this manuscript. Therefore, the reviewer would recommend the publication of this paper to Nature Communications after the following minor revision.

We thank the reviewer for the positive response and acknowledging the significance of our work for DRNNs. We also sincerely appreciate the reviewer’s suggestions to improve our manuscript by making more objective statements.

- What is the reason of much higher linearity of programming and erasing weights of the author’s device in comparison with other photonic synapses? This property is quite unique, but this was not completely demonstrated. This should be deeply covered in the manuscript.

- Under electronic erasing process, usually, other electrical erasing processes (application of voltage) have not shown this extremely linear modulation of conductivity like the author’s device [Adv. Funct. Mater. 2018, 28, 1804397; Nat. Commun. 2018, 9, 5106; Adv. Funct. Mater. 2019, 29, 1902374]. For responding to this question, the author showed Figure S9F, that is, erasing process with various erasing voltage amplitude. But, +30 and +10 V of erasing voltage did not make the device return to the initial points. Even though the previous studies [Adv. Funct. Mater. 2018, 28, 1804397; Nat. Commun. 2018, 9, 5106; Adv. Funct. Mater. 2019, 29, 1902374] completely returned to their initial states with erasing voltages, they did not show linear decrease under electronic erasing process. At the early stage, the conductivity was highly decreased and the amount of decreasing became smaller. Please the reason of higher linearity of the author’s device. This is very critical parts in this study.

We would like to address both the above comments in a single response because both the questions are related to the linearity of weight updates. Based on the reviewer’s input we have undertaken studies on the various factors that control the phenomena. We include additional data on newly fabricated Rhenium disulphide (ReS₂) and Poly(3-hexylthiophene) (P3HT) based systems. To demonstrate the universality of our approach, we have conducted further experiments on Molybdenum disulphide (MoS₂) and Black Phosphorus (BP) PENs, and have validated our methodology across these material systems as well.

As detailed in the first revision, we attribute the optical modulation due to persistent photoconductivity (PPC) to be the reason for linear programming of weights in our work. Upon illumination, the photogenerated holes are localized/trapped in states within the semiconductor bulk or/and at the semiconductor-dielectric interface resulting in a delayed recombination of the generated photocarriers. This results in a permanent change in the channel’s carrier concentration resulting in a non-volatile weight update. Carriers generated during the subsequent illumination adds to the existing carrier concentration with every input pulse and we are able to probe weights in a linear manner. We believe the *linear increase/decrease of weights in a material system depend on a number of factors including the nature of traps and its temporal response, the kinetics of photo-carrier generation and their recombination. We believe the programming scheme that we have adopted caters for many of these factors.*

Nature of defects:

The exact nature of traps or defects in a material system is dependent on the composition and the fabrication processes involved. In transition metal dichalcogenides, studies have indicated different possible origins for the electron trapping and detrapping mechanisms, including *surface adsorbates*, *electron trapping at the semiconductor-dielectric interface* and *intrinsic semiconductor defects/lattice defects* [2017. *Nature Communications*, 8(1), pp.1-8., 2013. *Nature Nanotechnology*, 8(7), pp.497-501., 2017. *Physical review letters*, 119(4), p.046101., Cho et al. 2013. *ACS Nano*, 7(9), pp.7751-7758., 2017. *npj 2D Materials and Applications*, 1(1), pp.1-9.]. Since our measurements were performed in high-vacuum conditions, the probability of surface adsorbates are highly reduced. Thus, we attribute the origin of trapping-detrapping mechanism in our measurements to defects in the semiconductor itself or/and the at the semiconductor-dielectric interface. The high value of subthreshold slope ($\sim 2.5\text{V}/\text{dec}$) in our devices support the presence of traps influencing the ReS₂ channel in accordance with similar measurements in 2013. *Nature Nanotechnology*, 8(7), pp.497-501. and 2018. *ACS Applied Materials & Interfaces*, 10(42), pp.36512-36522. In the present work, we utilize a combination of optical and electrical pulses to fill and empty these traps and to induce non-volatile conductance changes in our FETs. **The number of distinct conductance states are determined primarily by the programming pulse resolution and recombination kinetics of the photo-generated carriers, and hence, the programming pulses could be optimized to achieve a very good linearity.**

Programming scheme to maximize linearity:

To program states in a linear manner, we need to carefully select the gate voltage, the initial conductance state, the drain voltage and intensity of light illumination as explained below.

Firstly, to maximize linearity in write and erase, we apply and maintain a gate bias (-40V) to our devices to take it to their depleted state before starting our measurements (SI-Note-3 Figure-S5). The initial low conductance is a critical step in achieving linearity as also reported by other investigations (2019. *Science*, 364(6440), pp.570-574., 2019. *Advanced Functional Materials*, 29(31), p.1902374.). By constantly biasing the devices at a gate voltage that depletes the majority carriers in the system (-40V in the case of ReS₂ devices or negative V_{gs} for any n-type semiconductor), we ensure that the traps are empty and the background carrier concentration is minimized (2017. *Nature Communications*, 8(1), pp.1-8.). Upon photoexcitation, the applied gate voltage would increase the probability of photocarrier trapping (2017. *Nature Communications*, 8(1), pp.1-8.). Upon illumination, the photogenerated carriers fill up these traps, resulting in a permanent increase in the channel conductance due to PPC (also termed photogating in literature, e.g. 2013. *Nature Nanotechnology*, 8(7), pp.497-501. and 2018. *ACS Applied Materials & Interfaces*, 10(42), pp.36512-36522.). Finally, the V_{ds} should be high enough to overcome possible contact resistance effects and should result in an I_{ds} which is significantly larger than I_{gs} , necessary to ensure accurate weight update readouts.

In summary, a V_{gs} that depletes the major carriers and keeps the trap states empty, a V_{ds} that ensures the channel currents are much larger than the gate leakage currents, a low initial conductance/background carrier concentration to avoid interference with the photogenerated carriers and an appropriate light intensity to generate photocarriers which can respond to the applied V_{gs} to fill the traps should give the highest chance of linearity (Figure R1). This figure on our proposed mechanistic understanding is now added to the supporting information (SI-Note-7 Figure-S12).

Figure R1. Mechanistic understanding of the read, write and erase process.

We now present the data to support the assertions indicated above. We have also discussed this now in the supporting information extensively. Please refer to Notes-7 and 9. To prove the universal nature of this approach comprehensively, we have also performed additional analysis on freshly-prepared Molybdenum disulphide (MoS_2), Black Phosphorus (BP) and P3HT FETs as explained below.

Effect of constant gate bias V_{gs} :

We first show that the linearity is modulated as a function of the applied constant gate bias during optical potentiation for ReS_2 FETs. Figure R2 below shows the weight updates: optical potentiation with different constant gate biases, and electrical depression.

Figure R2. Weight changes in ReS_2 PENs [Data from 15 devices] as a function of the constant gate bias V_{gs} . For potentiation, blue light pulses of $\lambda=445\text{nm}$ and intensity= $65\text{mW}/\text{cm}^2$ was used with a pulse width and interval of 10s each. Key: Depleted- refers to a range of V_{gs} -40 to -50V depending on the V_{on} of the respective PEN. Similarly, Partially Depleted- refers to a range of V_{gs} -10 to +10V and Accumulated- refers to a range of V_{gs} +10 to +40V. The graph represents our conclusion from experimental measurement of 15 devices. The error bars represent the variation among devices.

As can be seen, maintaining a large negative V_{gs} that depletes the majority carriers in the system allows us to write weights more linearly when compared to partially depleted ($\sim 0\text{V}$) and accumulative (high $+V_{gs}$) voltages for ReS_2 FETs. As explained above, we believe this negative V_{gs} helps empty the traps, allowing us to control the photo-generated carriers more precisely, resulting in linear weight updates. Since we perturb the carrier concentration in small steps, the erasing is also fairly linear compared to the very high conductances reached at zero/positive V_{gs} . At zero/positive V_{gs} , our FET is already ON with a high concentration of background carrier in the channel (Figure S1F), which screens the effect of photo-generated carriers, resulting in lower magnitude and non-linear update of weights during potentiation. The larger value of absolute conductance also results in uncontrollable (non-linear) erase steps, sometimes even going below the level of initial conductance. The data presented in the above graph represent conclusions from 15 ReS_2 FETs. We have now conducted further experiments on freshly prepared MoS_2 and black phosphorus (BP) FETs and our conclusions remain valid as illustrated by Figure R3 below. *Please note that MoS_2 FETs are n-type and hence requires a large negative gate voltage to keeps the trap states empty, while p-type BP FETs require a high positive gate voltage for the same.*

Figure R3. Weight changes in (A) n-type MoS_2 [Data from 15 devices] and (B) p-type BP PENs [Data from 5 devices] as a function of the constant gate bias V_{gs} . For potentiation, blue light pulses of $\lambda=445\text{nm}$ and intensity= $65\text{mW}/\text{cm}^2$ was used with a pulse width and interval of 10s each. Key for MoS_2 : Depleted- refers to a range of V_{gs} -40 to -50V depending on the V_{on} of the respective PEN. Similarly, Partially Depleted- refers to a range of V_{gs} -10 to $+10\text{V}$ and Accumulated- refers to a range of V_{gs} $+10$ to $+40\text{V}$. Key for BP: Depleted- refers to a range of V_{gs} $+40$ to $+60\text{V}$ depending on the V_{on} of the respective PEN. Similarly, Partially Depleted- refers to a range of V_{gs} $+10$ to -10V and Accumulated- refers to a range of V_{gs} -40 to -60V . The graph represents our conclusion from experimental measurement of 15 MoS_2 and 5 BP FETs. The error bars represent the variation among devices.

Furthermore, we conducted additional experiments on P3HT-based FETs (Figure R4). Please note that the average retention of the non-volatile states in this configuration is $<200\text{s}$ as supposed to $>900\text{s}$ for our 2D TMDC FETs. However, the weight changes during optical potentiation still follows the same dependence on V_{gs} as explained above. At $V_{gs}=0\text{V}$, the P3HT FET is already ON (refer to Figure R6B below) and hence the linearity quickly ceases and reaches saturation. As expected, a large $+V_{gs}$ is necessary to empty the electron traps since P3HT is a p-type semiconductor and this results in improved linearity as shown below. This indicates that our approach is an universal methodology that can be applied to any semiconducting platform to extract better linearity.

Figure R4. Weight changes in p-type P3HT OFETs as a function of the constant gate bias V_{gs} . For potentiation, blue light pulses of $\lambda=445\text{nm}$ and intensity= $23\text{mW}/\text{cm}^2$ was used with a pulse width and interval of 2s each. Key for P3HT: Depleted- refers to a of V_{gs} $+60\text{V}$ and Partially Depleted- refers to a of V_{gs} 0V .

The effectiveness of the gate voltage in inducing write and erase linearity is also linked to the background conductance of the devices (2017. *Nature Communications*, 8(1), pp.1-8.). Medium and high initial conductance states result in lower magnitude of weight updates due to the screening effects of the background charge carrier concentration and the high absolute value of conductance results in uncontrollable non-linear asymmetric erasing (Figure R5).

Figure R5. Weight changes in (A) ReS₂ [Data from 15 devices] and (B) MoS₂ PENs [Data from 15 devices] as a function of the initial conductance state. Key: Low- refers to a range of 1-50nS depending on the respective PEN. Similarly, Medium- refers to a range 100nS-1µS and High- refers to a range of 1-100µS. The initial conductance here refers to the conductance of the FET before potentiation and depression measurements. For a fair comparison, the same fully depleting voltage ($V_{gs}=-40V$) was applied to all devices.

Effect of drain bias V_{ds} :

In our TMDC FETs, we typically apply a V_{ds} value of 0.1V for our measurements. Lower applied drain voltages resulted in inconsistent results especially when the gate leakage currents (I_{gs}) were high. The effect of V_{ds} can be more clearly demonstrated in the P3HT system by comparing the linearity of weight change responses under the same optical pulsing conditions and V_{gs} bias (Figure R6). As explained above, a V_{ds} that ensures the channel currents are much larger than the leakage currents is necessary for accurate weight read-outs. Interference of the gate leakage currents affect the read outs and their linearity. This is a very critical requirement as observed in 2016. *Nature Nanotechnology*, 11(9), p.769. and (2017. *Nature Communications*, 8(1), pp.1-8.). For the P3HT FETs, a high drain voltage of -80V is required to extract excellent linearly varying photo-memory effects. Low V_{ds} result in poor margins between I_{ds} and I_{gs} as evident from their transfer characteristics, impairing the extraction of linear optical weight updates. As shown in the figure below, at $V_{ds}=-1V$, the gate leakage current I_{gs} ($10^{-8}A$) is much larger than the drain current I_{ds} ($10^{-9}A$), preventing reliable weight readouts. At -5V, I_{ds} is larger than I_{gs} , but with a very low margin (<3x). At -60V and higher, I_{ds} is much larger than I_{gs} with good readout margins (>200x), allowing accurate weight readouts with very high signal to noise ratio. Please note that the magnitude of normalized weight changes at high drain voltages are lower than those at low drain voltages in the figure below. However, since the channel conductance at $V_{ds}=-5V$ is heavily influenced by the gate leakage currents (refer to the I_dV_g curves on the right), we believe that only the weights extracted from $V_{ds}=-80V$ reflect accurate read outs.

Figure R6. (A) Weight changes in p-type P3HT OFETs as a function of V_{ds} . For potentiation, blue light pulses of $\lambda=445nm$ and intensity= $23mW/cm^2$ was used with a pulse width and interval of 2s each. V_{gs} was held constantly at $+60V$ (fully depleted mode) throughout the measurements. (B) Transfer characteristics of P3HT OFETs under different V_{ds} .

Effect of optical illumination (Intensity):

We next show that the weight update steps could be further modulated by changing the intensity of illumination (Figure R7). Higher optical intensities (consequently increased photocarrier trapping) results in increased weight changes, albeit with an increased spread. Similar phenomenon is also noted in newly tested MoS_2 FETs.

Figure R7. Weight changes as a function of the light intensity for $\lambda=$ (A) 623, (B) 525 and (C) 445nm, 10s ON, 10s OFF in ReS₂ PENs [Data from 5 devices]. (D) Weight changes as a function of the light intensity for $\lambda=$ 445nm, 10s ON, 10s OFF in MoS₂ PENs [Data from 5 devices]. The graph represents our conclusion from experimental measurement of 5 ReS₂ and 5 MoS₂ FETs. The error bars represent the variation among devices.

Linearity of electrical “erase”:

The “erase” process for n-type ReS₂ via application of a positive gate voltage raises the Fermi level to induce electron accumulation in the channel and accelerates the recombination of electrons with the holes in the trap sites resulting in lowered current levels as shown in Figures-2B-D and Supporting Information Figure-S9. This is in line with the conclusions of Robertson, J., et al. in 2012. *Nature Materials*, 11(4), pp.301-305. and 2013. *Nature Nanotechnology*, 8(7), pp.497-501 where voltage pulses that induced majority carriers were used to eliminate persistent photocurrent. With regards to the linear nature of erase, we would like to clarify that this is made possible by carefully selecting the erase voltages.

“... For responding to this question, the author showed Figure S9F, that is, erasing process with various erasing voltage amplitude. But, +30 and +10 V of erasing voltage did not make the device return to the initial points...”

First, we would like to clarify that the data shown in our previous version as Figure-S9F (shown below as Figure B1) was presented just to demonstrate that weight updates during erase could be modulated widely by choosing the correct voltage level. The optical illumination intensity utilized to demonstrate linear potentiation in Figure-S9F was 230mW/cm² as opposed to the optimized 65mW/cm² in the rest of the manuscript. As a result of the large magnitude of potentiation with strong illumination (230mW/cm²), we were unable to program the states in the depression phase back to the initial state.

Figure B1.

However, under our optimized programming schemes as detailed in Figures 2B-D, S9B-D and in the additional data collected for this response, we are able to program the states in the depression phase back to the initial state. However, this requires extensive optimization as shown below.

Figure R8. Weight changes in ReS₂ PENs [Data from 15 devices] as a function of the erase voltages. (A-C) depicts the optimization of erase voltages as a function of the optical illumination wavelength. The graph represents our conclusion from experimental measurements of 15 ReS₂ FETs. The error bars represent the variation among devices.

Increasing the erase gate voltage raises the Fermi level further to accumulate more electrons in the channel which can recombine with the trapped holes, allowing us to tune our symmetry. This is similar to strategies used by other groups to achieve desired conductance levels [2018. *Science Advances*, 4(4), p.eaap7916 and 2015. *Science Advances*, 1(9), p.e1500640]. The input pulses can also be modified as per the device's response to maximize linearity as adopted in 2018. *Nat. Communications*, 9, 5106, 2016. *Nanotechnology*, 27(36), p.365204., 2015, *IEEE/ACM International Conference on Computer-Aided Design (ICCAD)* (pp. 194-199). IEEE., 2017. *IEEE Electron Device Letters*, 38(8), pp.1023-1026. As indicated from the figures, higher magnitude of absolute conductance requires larger amplitude of erase voltages to erase the channel conductance back to the initial state and achieve symmetry. Intuitively, the large magnitude of absolute and percentage weight changes during potentiation reflects the larger number of trapped holes and hence, the larger amplitude of erase voltages adopted in turn indicates the magnitude by which the Fermi level should be raised in order to accumulate enough additional electrons in the channel to recombine with larger number of trapped holes.

In summary, the exquisite write linearity afforded by the optical gating is the major phenomenon we exploit in this work to get very accurate weights in a DRNN. We have now included this mechanistic discussion in the main text (Please refer to Page 11) and in much greater detail in the Supporting Information Notes-7 and 9.

Comparison to works highlighted by the reviewer:

As explained in the first revision, we would like to re-emphasize that although persistent photoconductivity is widely observed across several materials ranging from organic semiconductors, metal oxide semiconductors, III-V semiconductors and 2D TMDC semiconductors, the extent and temporal dynamics of this phenomenon varies widely as a function of the background carrier concentration, illumination parameters, nature of defects in the semiconductor and dielectric, selection of contacts and even substrate cleaning methods. Hence, comparison across reports is very challenging. However as per the reviewer's suggestion, we now have attempted to provide a detailed comparison of our work to the referred manuscripts based on the experimental evidence presented in these works and some of our assumptions of their testing conditions.

In *Adv. Funct. Mater.* 2018, 28, 1804397, the authors present a three-terminal device with IGZO and the ion gel, both prepared by a solution process, used as the semiconducting channel and blocking gate dielectric layer, respectively. GO nanosheets modified with octadecyl alkyl chains (alkylated GO) was deposited on the IGZO channel and capped by an ion-gel dielectric. The GO nanosheets and the long alkyl chains served as charge-trapping sites and a nanoscale tunneling gate dielectric, respectively. The ion gel consisted of poly(vinylidene difluoride-cohexafluoropropylene) [P(VDF-HFP)] and 1-ethyl-3-methylimidazolium bis(trifluoromethylsulfonyl)imide [(EMIM)(TFSI)] in a weight ratio of 1:4. The authors indicate that LTP-LTD curves are achieved via inputs of 2 types- voltage-only and mixed (voltage+light) pulses. The authors indicate that the application of the appropriate gate voltage allows carriers to be trapped and detrapped from the device, allowing for conductivity modulation.

We would like to first point to some critical differences in the material systems employed- namely an ionic gel which sets up strong electric fields in the electrical double layer (EDL) (in contrast to conventional SiO₂ employed in our studies) as well as the large bandgap oxide semiconductor (which limits photosensitivity as compared to our TMDCs). Since the alkylated GO is discontinuous (AFM images in supporting information of manuscript), the ionic gel is also directly in contact with the IGZO layer. Thus we believe, the application of gate voltages results in two processes occurring at the same time- trapping/ detrapping of electrons from the GO as well as accumulation/ depletion of carriers from IGZO. However the authors do not explicitly discuss these two processes and focus primarily on the GO based electrical processes. We believe that the limits of linearity achieved via voltage-only pulses in this system is caused by the 2 carrier contributing mechanisms underplay- charge detrapping from the floating gate and carrier depletion due to the strong EDL capacitive effects in the ion gel, thus making it difficult to precisely control the step size of weight updates in a linear manner. Similarly in the depression phase, positive voltages would inject electrons into the GO, but at the same time induce carrier accumulation in IGZO due to the EDL effects, once again resulting in a non-linear control of weight updates. For the mixed input pulses, simultaneous application of voltage and light pulses clearly create LTP and LTD as evidenced by Figure 3c. In this mode, the light pulses cause photogenerated holes in the valence band of IGZO to further detrapp electrons from the GO nanosheets, resulting in higher weight changes with respect to the voltage-only mode. However, the linearity could again be compromised because of the 4 independent effects occurring during this pulsing scheme- detrapping of electrons from the floating gate, lowering of Fermi level of IGZO and hole trapping upon illumination, and counteracting EDL effects of the ion gel. This is supported by the worsened linearity metric extracted from the mixed input curves (value of 4.24/8 when compared to 5/8 for voltage-only inputs) and Figure 3f. Finally, the optical signals employed in this system doesn't seem to change the trapping dynamics of the system which is again different from our results, once again pointing to the difference in the nature of traps in both systems. Figures S13 and S14 of the referenced manuscript shows much better linearity with respect to Figure 3D but since the exact pulsing conditions and the device parameters are not clearly mentioned, we cannot judge more at this moment. Another important factor to note is the short duration of voltage and light pulses utilised in the study (ms time scale) which may also not allow for complete filling and emptying of the traps. In general for a given illumination condition, the lower bandgap and ultrathin dimensions of our TMDC FETs will allow higher probability for complete filling and emptying of the traps when compared to the high band gap oxide semiconductors. The ion gel response may also have a frequency response (due to redistribution of the ions within them) that convolutes these trap filling and emptying processes.

One additional factor that could be occurring in this system is the possibility of oxygen vacancy generation under high electric fields that has been noted in IGZO interfaced with EMIM-TFSI (2019. *Small*, 15(27), p.1901457.).

In comparison, we utilize light-only pulses combined with a constant voltage-bias for programming non-volatile memory states as explained earlier. The low band gap of our TMDC FETs allows us to program non-volatile states without additional voltage pulses in comparison to high band gap oxide semiconductors. The constant negative electrical bias helps to empty the trap states and the optical pulses generates carriers to fill these states, in turn allowing us to perturb the channel conductivity with higher precision, resulting in better linearity. Also, we do not have multiple coupled charge trapping-detrapping mechanisms contributing to the weight updates like in the referred manuscript, again resulting in better control over the weight updates.

In short, the nature of traps are very clearly different in both the systems. While the referred manuscript employs multiple charge modulation phenomena (voltage and light-assisted hole trapping and detrapping from a floating gate; strong capacitive EDL effects), we utilize the generation and recombination of photogenerated carriers to achieve weight updates with better linearity.

In *Adv. Funct. Mater.* 2019, 29, 1902374, a combination of light-only and voltage-only programming pulses are used for potentiation and depression respectively. A charge trapping layer composed of carbon dots embedded in a silk protein matrix traps photogenerated electrons and transfers holes to the pentacene semiconducting channel, resulting in higher p-conductivity or potentiation during optical stimulation. Application of negative voltage to the pentacene results in injection of holes to the floating gate, reducing the conductance or depression. We believe that the reviewer is pointing to Figure 3e as a comparison to our LTP-LTD curves. We would like to first point out that it is not clear in Figure 3e whether the data presented is STP and STD and not LTP or LTD. The input pulse used during the potentiation in Figure 3e is $0.15\text{mW}/\text{cm}^2$, pulse width=0.5s. As per the author's statement, the device depicts STP for optical pulses of duration 1s or less and LTP under the application of input optical pulse at 3, 5, and 10 s duration. This is corroborated by Figures 3f-h. Similarly for depression, -10V pulses of pulse width 0.05s induce STD (not LTD) as shown in Figure 3d.

Based on the information provided, we believe the differences in write and erase linearity could be traced to the differences during programming. Unlike our case where the devices are held in deep depletion during the optical programming process, there is no gate voltage applied during the photopotential process. The depleting voltage not only reduces the background carrier concentrations, but can also drive the photoexcited minority carriers to stable deep trap states. In the case of the pentacene device, the lack of applied voltage during the photoexcitation could imply that the photogenerated minority carriers occupy shallow trap states, which in turn during the electrical erasing process results in non-linear behaviour.

In *Nat. Commun.* 2018, 9, 5106, the conductance changes in a photo-resistor (sensor) upon illumination induce weight changes in a floating gate transistor (synaptic device) connected in series. Here the LTP-LTD weight updates (defined as a conductance change $>0.3\%$ with a retention of 8 seconds) in the synaptic transistor are not directly controlled by optically inputs (like in our case) but indirectly controlled by the resistance drop across a photo-resistor which acts as a load sensor. The weight changes in the synaptic device as such are caused by the trapping and de-trapping of electrons in the weight control layer (WCL) [oxidized boron] on h-BN. The light-induced drop in the photoresistance of the optical sensing device, only creates additional V_{ds} drop, changing the conductance levels of the synaptic device. The large number of linearly addressable states (~ 600) achieved in this work is a result of the linearly varying load photo-resistor (termed as optical sensing device in the referred manuscript), causing analogous voltage drop across the source-drain electrodes of the synaptic device. LTP and LTD is already achieved in the dark condition and the optical signals just add voltage drop across the synaptic transistor

without disturbing the charge trapping dynamics of the synaptic device. This is clear from the fact that the non-linearity factor does not change with light inputs.

In short, this work portrays an electrical LTP and LTD of the synaptic device with a photoresistor in series. This is very different from our approach where we fuse the sensor and synapse into 1 device. More critically, the optical signals directly program conductance states in our device. When we adopt a pure-electrical programming scheme in our devices similar to the referred manuscript, the LTP-LTD we get are much worse than this report. Both linearity and symmetry are affected and potentiation dominates depression as can be seen from the Figure R9 below. This again indicates the differences in trapping dynamics in the systems under comparison.

Figure R9. Pure-electrical weight changes in ReS₂ PENs. Potentiation is achieved via voltage pulses of -40V and depression using +40V. Other pulsing parameters remain constant [pulse width=1s, pulse interval=0.1s, number=10].

In short, the nature of traps and the methodology of weight updates are very clearly different in both the systems. While the referred manuscript achieves weight updates with very good linearity with voltage pulses and utilizes light pulses only to shift the absolute value of weights, we utilize the generation and recombination of photogenerated carriers to achieve weight updates with good linearity. While the referred manuscript employs 2 devices- 1 sensor + 1 synapse to achieve this, our device serves as both the sensor and synapse.

Thus to briefly summarise the comparison to these three reports, the better linearity we achieve could be attributed to the biasing and testing conditions of our TMDC FETs to extract maximum linearity from our systems. A V_{gs} that keeps the trap states empty and subsequently drives the photogenerated carriers into the traps, a V_{ds} that ensures the leakage currents do not interfere with the weight readouts, a low initial background carrier concentration to prevent screening effects and a light intensity sufficient to fill the traps ensures us the highest chance of linearity. We have now added this discussion on mechanistic comparison in both the main text and supporting information. Please refer to pages 11 and 15 in the main text and SI-Notes-7 and 9 in the supporting information.

Next, we compare the referred neural network implementations with our approach. In *Adv. Funct. Mater.* **2018, 28, 1804397**, the authors have achieved up to 200 weight states and performed supervised MNIST digit classification using a single layer neural network with 7850 parameters. In *Adv. Funct. Mater.* **2019, 29, 1902374**, the authors also achieved similar number of states and implemented a neural network with similar number of parameters. The authors in *Nat. Commun.* **2018, 9, 5106** achieved up to 600 weight states and colored or mixed color digit recognition were performed using a neural network with ~4.7k parameters. While our proposed device achieves similar number of states (600) to *Nat. Commun.* **2018, 9, 5106**, the proposed deep neural network consists of ~3.6M parameters, orders of magnitude higher than these prior works. Moreover, while the authors in these papers use hardware based online backpropagation learning, our offline training methodology enables us to use more sophisticated

momentum based learning rules such as ADAM. Several algorithmic modifications such as inclusion of momentum term, per-parameter basis adaptation of learning parameters, bias-corrected parameter estimation etc. makes ADAM very suitable for deep multilayered neural networks with complex and non-stationary learning objectives, resulting in its growing popularity and success in computer science community in recent years [Supporting Information Note-8].

Finally, our proposed two-shot writing scheme provides two key advantages over existing weight update schemes. Firstly, in some of the existing literature, device-to-device variability is ignored for neural network simulations. For large networks neural with millions of parameters, we need to account for device-to-device variability for realistic simulations. Secondly, while taking multiple measurements and getting the best fit curve for each device results in more accurate weight updates, it becomes impractical for deep networks with very large number of weights. The proposed two-shot writing scheme that exploits our exceptional linearity in updates not only accounts for device variability, but also provides a much simpler alternative using only one additional measurement pulse, at little penalty on performance [Supporting Information Note-7 Figures-S18, S21]. The scalable weight update scheme and linearity required to implement different types of layers (convolutional, recurrent, fully connected etc.) [Figure-3C] makes our approach an ideal candidate for implementing truly deep networks for various complex cognitive tasks.

- Please add the full name of TMDC (title).

As per the reviewer's suggestion, we have now changed the title to "*Optogenetics-Inspired Light-Driven 2D Transition Metal Dichalcogenide Computational Circuits Enable In-Memory Computing for Deep Recurrent Neural Networks*".

- In the response letter, why did the author exclude Intrinsic semiconductor defects as the origin of trapping-detrapping mechanism? Explain this.

We would like to clarify that we **do not exclude** intrinsic semiconductor defects as the origin of trapping-detrapping mechanism. In the response letter, we have indicated different possible origins for the electron trapping and detrapping mechanisms based on literature and our observations. Commonly attributed sources of traps include *surface adsorbates* like gas molecules (e.g. oxygen and water) from the environment, *traps at the semiconductor-dielectric interface* (e.g. dangling Si-O bonds) and *intrinsic semiconductor defects* (e.g. S and Re vacancies) as explained in detail in the first response letter.

Since our measurements were performed in high-vacuum conditions, the probability of surface adsorbates are highly reduced. **Thus, we attribute the origin of trapping-detrapping mechanism in our measurements to defects in the semiconductor itself and the at the semiconductor-dielectric interface.** The high value of subthreshold slope ($\sim 2.5\text{V/dec}$) in our devices support the presence of traps influencing the ReS_2 channel in accordance with similar measurements in 2013. *Nature Nanotechnology*, 8(7), pp.497-501. and 2018. *ACS Applied Materials & Interfaces*, 10(42), pp.36512-36522. Recent studies have revealed the possibility of sulphur vacancies [2017. *Nature Communications*, 8(1), pp.1-8.] and transition metal vacancies [2017. *Physical review letters*, 119(4), p.046101.] to be the source of traps that can significantly affect the optoelectronic properties of 2D TMDCs. In the present work, we utilize a combination of optical and electrical pulses to fill and empty these traps and hence, induce non-volatile conductance changes in our FETs.

Reviewer #3 (Remarks to the Author):

Authors have made some attempts to answer the questions. However, I still think that the manuscript would be better suited for a more specialized journal. I also still do not think that there is any optogenetics motivation here. Showing optical write and electrical erase in one device is easy. But accomplishing the same in a large array is simply impractical.

We thank the reviewer for his/her critical and valuable comments. We agree with the reviewer that demonstrating optical write and electrical erase on one device is easy and implementing a large array would have its challenges. However, new, scalable and reproducible growth methods in TMDCs continue to be developed (2019. *Science Advances*, 5(7), p.eaaw3180., 2018. *Nature Materials*, 17(4), pp.318-322., 2018. *Science*, 362(6415), pp.665-670., 2018. *Nature*, 556(7701), pp.355-359.). Numerous demonstrations of large scale TFT arrays based on TMDCs with photodetection also points to the possibility that such neuromorphic systems could be feasible. Through significant progress in the wafer-scale growth of 2D materials and heterointegration, the realization of large arrays of neuromorphic elements based on TMDCs might not be too farfetched. The portability of our demonstrated concepts to other semiconductor systems also point to alternative routes for the concepts to be realised. We would like to point out that our manuscript demonstrates several novelties, which from a materials development and device concept perspective are first of its kind.

1. The final application we address, that of deep recurrent neural networks (DRNNs) has not been tackled before with memristive devices. Most reports demonstrate memristive devices with limited precision weights that can only be used for shallow feed-forward networks trained to classify simple datasets. These devices and platforms are incapable of addressing applications like speech recognition and natural language processing (NLP) that require learning of temporal signals. We exploit the exquisite write linearity afforded by the optical gating to get very accurate weights, demanded by DRNNs- first demonstration of its kind.

2. We comprehensively demonstrate the unparalleled linear switching ability in our systems through a number of characterizations. We rigorously analyse the signal-to-noise ratio, linear dynamic range and various other metrics to benchmark our performance with the state-of-the-art electrical memristive systems. While programming with optical pulses have been demonstrated before (including our own previous work), such a rigorous analysis on optical programming schemes has not been performed before to the best of our knowledge. The value reported in the manuscript (980 linearly addressable states) exceed the state-of-the-art as shown in Figure 3D. As can be seen, we go beyond both state-of-the-art pure-electrical and pure-optical implementations, reiterating the significance of our contribution to the field of neuromorphic computing.

4. We have demonstrated unique features of wavelength selectivity and have now added intensity dependence in this revision, again measured across multiple devices and materials. We have performed additional experiments to prove the selective nature of our optoelectronic programming scheme. We have now demonstrated a 36 element array consisting of 16 light-sensitive 2D TMDC ReS₂ and 20 light-insensitive oxide semiconductor FETs and validated selective probing of light-sensitive ReS₂ PENs. All these experiments are first of its kind and have not been demonstrated before.

5. To prove the universal nature of the proposed concept, we have carried out additional experiments on Molybdenum disulphide (MoS₂), Black Phosphorus (BP) and Poly(3-hexylthiophene) (P3HT) semiconducting films and proved the same approach to be effective across multiple material systems. Never before has such an extensive study been performed in one single work.

6. We have tested the limits of linearity in our systems with additional experiments and observed the exact saturation level is not important as long as the dynamic range is high enough. The 2-shot writing scheme we adopt takes into account the more critical variation in the slope of the curve to eliminate any sort of mismatches at the network level. The proposed two-shot writing scheme provides two key advantages over existing weight update schemes. Firstly, we take into account device-to-device variability for realistic simulations- often ignored in many works. Secondly, the proposed scheme provides a much simpler alternative for accurate weight updates using only one additional measurement pulse at little penalty on performance, in comparison to taking multiple measurements and getting the best fit curve for each device- impractical for deep networks with very large number of weights. The offline training methodology we adopt enables us to use more sophisticated momentum based learning rules such as ADAM, which takes into account several algorithmic modifications such as inclusion of momentum term, per-parameter basis adaptation of learning parameters, and bias-corrected parameter estimation etc., ideal for deep multi-layered neural networks with complex and non-stationary learning objectives. Thus, we advance the field not only from a materials and device viewpoint, but also from the algorithm perspective.

We firmly believe that the proof-of-concept experiments we demonstrate goes well beyond the state-of-the-art, will be of interest to a broad scientific audience in materials science, computing, applied physics and electrical engineering, and warrants publication in a reputed journal like Nature Communications.

REVIEWERS' COMMENTS:

Reviewer #1 (Remarks to the Author):

The authors have addressed my previous questions in the revision.

Reviewer #2 (Remarks to the Author):

In the manuscript titled "Optogenetics-Inspired Light-Driven 2D Transition Metal Dichalcogenide Computational Circuits Enable In-Memory Computing for Deep Recurrent Neural Networks", the authors investigated a unique neuromorphic computing platform comprising of optoelectronically-modulated artificial neurons and synapses with selective excitability and linear conductance changes, sufficing blind weight-update protocols for highly-parallel DRNNs. In addition, in-memory computations across 980 addressable states with an average signal-to-noise ratio as high as 77 utilizing photo-excitabile neuristors was demonstrated and this extremely high linear dynamic range was exploited to simulate a memristive DRNN to recognize spoken commands with >90% accuracy. The reviewer thinks that researchers in this field would be very interested in the contents of this manuscript.

Thus, the reviewer recommends recent references about 2D material-based optoelectronic artificial synapses [ACS Nano 2020, 14, 1, 746-754; Adv. Mater. 32, 1906899] in the introduction part. After this, the reviewer thinks that the manuscript can be accepted by Nature Communications, because it was largely strengthened after the revision.

REVIEWERS' COMMENTS:

Reviewer #1 (Remarks to the Author):

The authors have addressed my previous questions in the revision.

We thank the reviewer for the positive feedback on our revised manuscript.

Reviewer #2 (Remarks to the Author):

In the manuscript titled “Optogenetics-Inspired Light-Driven 2D Transition Metal Dichalcogenide Computational Circuits Enable In-Memory Computing for Deep Recurrent Neural Networks”, the authors investigated a unique neuromorphic computing platform comprising of optoelectronically-modulated artificial neurons and synapses with selective excitability and linear conductance changes, sufficing blind weight-update protocols for highly-parallel DRNNs. In addition, in-memory computations across 980 addressable states with an average signal-to-noise ratio as high as 77 utilizing photo-excitabile neuristors was demonstrated and this extremely high linear dynamic range was exploited to simulate a memristive DRNN to recognize spoken commands with >90% accuracy. The reviewer thinks that researchers in this field would be very interested in the contents of this manuscript. Thus, the reviewer recommends recent references about 2D material-based optoelectronic artificial synapses [ACS Nano 2020, 14, 1, 746-754; Adv. Mater. 32, 1906899] in the introduction part. After this, the reviewer thinks that the manuscript can be accepted by Nature Communications, because it was largely strengthened after the revision.

We thank the referee for the positive comments on our work and the revision. We have now added the 2 recent references suggested by the referee.